# ON THE GEOMETRY OF DEEP BAYESIAN ACTIVE LEARNING

## ABSTRACT

We present geometric Bayesian active learning by disagreements (GBALD), a framework that performs BALD on its geometric interpretation interacting with a deep learning model. There are two main components in GBALD: initial acquisitions based on core-set construction and model uncertainty estimation with those initial acquisitions. Our **key innovation** is to construct the core-set on an ellipsoid, not typical sphere, preventing its updates towards the boundary regions of the distributions. **Main improvements** over BALD are twofold: relieving sensitivity to uninformative prior and reducing redundant information of model uncertainty. To guarantee the improvements, our generalization analysis proves that, compared to typical Bayesian spherical interpretation, geodesic search with ellipsoid can derive a tighter lower error bound and achieve higher probability to obtain a nearly zero error. Experiments on acquisitions with several scenarios demonstrate that, yielding slight perturbations to noisy and repeated samples, GBALD further achieves significant accuracy improvements than BALD, BatchBALD and other baselines.

## 1 INTRODUCTION

Lack of training labels restricts the performance of deep neural networks (DNNs), though prices of GPU resources were falling fast. Recently, leveraging the abundance of unlabeled data has become a potential solution to relieve this bottleneck whereby expert knowledge is involved to annotate those unlabeled data. In such setting, the deep learning community introduced **active learning (AL)** (Gal et al., 2017) that, maximizing the model uncertainty (Ashukha et al., 2019; Lakshminarayanan et al., 2017) to acquire a set of highly informative or representative unlabeled data, and solicit experts' annotations. During this AL process, the learning model tries to achieve a desired accuracy using minimal data labeling. Recent shift of model uncertainty in many fields, such as Bayesian neural networks (Blundell et al., 2015), Monte-Carlo (MC) dropout (Gal & Ghahramani, 2016), and Bayesian core-set construction (Sener & Savarese, 2018), shows that, new scenarios arise from deep Bayesian AL (Pinsler et al., 2019; Kirsch et al., 2019).

**Bayesian AL** (Golovin et al., 2010; Jedoui et al., 2019) presents an expressive probabilistic interpretation on model uncertainty (Gal & Ghahramani, 2016). Theoretically, for a simple regression model such as linear, logistic, and probit, AL can derive their closed-forms on updating one sparse subset that maximally reduces the uncertainty of the posteriors over the regression parameters (Pinsler et al., 2019). However, for a DNN model, optimizing massive training parameters is not easily tractable. It is thus that Bayesian approximation provides alternatives including importance sampling (Doucet et al., 2000) and Frank-Wolfe optimization (Vavasis, 1992). With importance sampling, a typical approach is to express the information gain in terms of the predictive entropy over the model, and it is called Bayesian active learning by disagreements (BALD) (Houlsby et al., 2011).

**BALD** has two interpretations: model uncertainty estimation and core-set construction. To estimate the model uncertainty, a greedy strategy is applied to select those data that maximize the parameter disagreements between the current training model and its subsequent updates as (Gal et al., 2017). However, naively interacting with BALD using uninformative prior (Strachan & Van Dijk, 2003)(Price & Manson, 2002), which can be created to reflect a balance among outcomes when no information is available, leads to unstable biased acquisitions (Gao et al., 2020), e.g. insufficient prior labels. Moreover, the similarity or consistency of those acquisitions to the previous acquired samples, brings redundant information to the model and decelerates its training.

**Core-set construction** (Campbell & Broderick, 2018) avoids the greedy interaction to the model by capturing characteristics of the data distributions. By modeling the complete data posterior over the

distributions of parameters, BALD can be deemed as a core-set construction process on a sphere (Kirsch et al., 2019), which seamlessly solicits a compact subset to approximate the input data distribution, and efficiently mitigates the sensitivity to uninformative prior and redundant information.

From the view of geometry, updates of core-set construction is usually optimized with sphere geodesic as (Nie et al., 2013; Wang et al., 2019). Once the core-set is obtained, deep AL immediately seeks annotations from experts and starts the training. However, data points located at the boundary regions of the distribution, usually win uniform distribution, cannot be highly-representative candidates for the core-set. Therefore, constructing the core-set on a sphere may not be the optimal choice for deep AL.

This paper presents a novel AL framework, namely **Geometric BALD (GBALD)**, over the geometric interpretation of BALD that, interpreting BALD with core-set construction on an ellipsoid, initializes an effective representation to drive a DNN model. The goal is to seek for significant

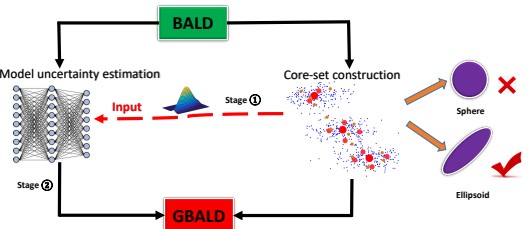

Figure 1: Illustration of two-stage GBALD framework. BALD has two types of interpretation: model uncertainty estimation and core-set construction where the deeper the color of the core-set element, the higher the representation; GBALD integrates them into a uniform framework. Stage ①: core-set construction is with an ellipsoid, not typical sphere, representing the original distribution to initialize the input features of DNN. Stage ②: model uncertainty estimation with those initial acquisitions then derives highly informative and representative samples for DNN.

accuracy improvements against an uninformative prior and redundant information. Figure 1 describes this two-stage framework. In the first stage, geometric core-set construction on an ellipsoid initializes effective acquisitions to start a DNN model regardless of the uninformative prior. Taking the core-set as the input features, the next stage ranks the batch acquisitions of model uncertainty according to their geometric representativeness, and then solicits some highly-representative examples from the batch. With the representation constraints, the ranked acquisitions reduce the probability of sampling nearby samples of the previous acquisitions, preventing redundant acquisitions. To guarantee the improvement, our generalization analysis shows that, the lower bound of generalization errors of AL with the ellipsoid is proven to be tighter than that of AL with the sphere. Achieving a nearly zero generalization error by AL with ellipsoid is also proven to have higher probability. Contributions of this paper can be summarized from Geometric, Algorithmic, and Theoretical perspectives.

- Geometrically, our key innovation is to construct the core-set on an ellipsoid, not typical sphere, preventing its updates towards the boundary regions of the distributions.

- In term of algorithm design, in our work, from a Bayesian perspective, we propose a two-stage framework that sequentially introduces the core-set representation and model uncertainty, strengthening their performance "independently". Moreover, different to the typical BALD optimizations, we present geometric solvers to construct core-set and estimate model uncertainty, which result in a different view for Bayesian active learning.

- Theoretically, to guarantee those improvements, our generalization analysis proves that, compared to typical Bayesian spherical interpretation, geodesic search with ellipsoid can derive a tighter lower error bound and achieve higher probability to obtain a nearly zero error. See Appendix B.

The rest of this paper is organized as follows. In Section 2, we first review the related work. Secondly, we elaborate BALD and GBALD in Sections 3 and 4, respectively. Experimental results are presented in Section 5. Finally, we conclude this paper in Section 6.

## 2 RELATED WORK

**Model uncertainty.** In deep learning community, AL (Cohn et al., 1994) was introduced to improve the training of a DNN model by annotating unlabeled data, where the data which maximize the model uncertainty (Lakshminarayanan et al., 2017) are the primary acquisitions. For example, in ensemble deep learning (Ashukha et al., 2019), out-of-domain uncertainty estimation selects those data which do not follow the same distribution as the input training data; in-domain uncertainty draws the data from the original input distribution, producing reliable probability estimates. Gal &

Ghahramani (2016) use MC dropout to estimate predictive uncertainty for approximating a Bayesian convolutional neural network. Lakshminarayanan et al. (2017) estimate predictive uncertainty using a proper scoring rule as the training criteria to fed a DNN.

**Bayesian AL.** Taking a Bayesian perspective (Golovin et al., 2010), AL can be deemed as minimizing the Bayesian posterior risk with multiple label acquisitions over the input unlabeled data. A potential informative approach is to reduce the uncertainty about the parameters using Shannon's entropy (Tang et al., 2002). This can be interpreted as seeking the acquisitions for which the Bayesian parameters under the posterior disagree about the outcome the most, so this acquisition algorithm is referred to as Bayesian active learning by disagreement (BALD) (Houlsby et al., 2011).

**Deep AL.** Recently, deep Bayesian AL attracted our eyes. Gal et al. (2017) proposed to cooperate BALD with a DNN to improve the training. The unlabeled data which maximize the model uncertainty provide positive feedback. However, it needs to repeatedly update the model until the acquisition budget is exhausted. To improve the acquisition efficiency, batch sampling with BALD is applied as (Kirsch et al., 2019; Pinsler et al., 2019). In BatchBALD, Kirsch et al. (2019) developed a tractable approximation to the mutual information of one batch of unlabeled data and current model parameters. However, those uncertainty evaluations of Bayesian AL whether in single or batch acquisitions all take greedy strategies, which lead to computationally infeasible, or excursive parameter estimations. For deep Bayesian AL, being short of interactions to DNN can not maximally drive their model performance as (Pinsler et al., 2019; Sener & Savarese, 2018), etc.

## 3 BALD

BALD has two different interpretations: model uncertainty estimation and core-set construction. We simply introduce them in this section.

### 3.1 MODEL UNCERTAINTY ESTIMATION

We consider a discriminative model $p(y|x,\theta)$ parameterized by $\theta$ that maps $x \in \mathcal{X}$ into an output distribution over a set of $y \in \mathcal{Y}$. Given an initial labeled (training) set $\mathcal{D}_0 \in \mathcal{X} \times \mathcal{Y}$, the Bayesian inference over this parameterized model is to estimate the posterior $p(\theta|\mathcal{D}_0)$, i.e. estimate $\theta$ by repeatedly updating $\mathcal{D}_0$. AL adopts this setting from a Bayesian view.

With AL, the learner can choose unlabeled data from $\mathcal{D}_u = \{x_i\}_{j=1}^N \in \mathcal{X}$, to observe the outputs of the current model, maximizing the uncertainty of the model parameters. Houlsby et al. (2011) proposed a greedy strategy termed BALD to update $\mathcal{D}_0$ by estimating a desired data $x^*$ that maximizes the decrease in expected posterior entropy:

$$x^* = \arg\max_{x \in \mathcal{D}_u} \mathrm{H}[\theta|\mathcal{D}_0] - \mathbb{E}_{y \sim p(y|x,\mathcal{D}_0)}\Big[\mathrm{H}[\theta|x,y,\mathcal{D}_0]\Big], \qquad (1)$$

where the labeled and unlabeled sets are updated by $\mathcal{D}_0 = \mathcal{D}_0 \cup \{x^*,y^*\}, \mathcal{D}_u = \mathcal{D}_u \backslash x^*$, and $y^*$ denotes the output of $x^*$. In deep AL, $y^*$ can be annotated as a label from experts and $\theta$ yields a DNN model.

### 3.2 CORE-SET CONSTRUCTION

Let $p(\theta|\mathcal{D}_0)$ be updated by its log posterior $\log p(\theta|\mathcal{D}_0, x^*)$, $y^* \in \{y_i\}_{i=1}^N$, assume the outputs are conditional independent of the inputs, i.e. $p(y^*|x^*, D_0) = \int_\theta p(y^*|x^*,\theta)p(\theta|D_0)\mathrm{d}\theta$, then we have the *complete data log posterior following* (Pinsler et al., 2019):

$$\mathbb{E}_{y^*}[\log p(\theta|\mathcal{D}_0, x^*, y^*)] = \mathbb{E}_{y^*}[\log p(\theta|\mathcal{D}_0) + \log p(y^*|x^*,\theta) - \log p(y^*|x^*,\mathcal{D}_0)]$$

$$= \log p(\theta|\mathcal{D}_0) + \mathbb{E}_{y^*}\Big[\log p(y^*|x^*,\theta) + \mathrm{H}[y^*|x^*,\mathcal{D}_0]\Big]$$

$$= \log p(\theta|\mathcal{D}_0) + \sum_{i=1}^N \left(\mathbb{E}_{y_i}\Big[\log p(y_i|x_i,\theta) + \mathrm{H}[y_i|x_i,\mathcal{D}_0]\Big]\right). \qquad (2)$$

The key idea of core-set construction is to approximate the log posterior of Eq. (2) by a subset of $D'_u \subseteq D_u$ such that: $\mathbb{E}_{\mathcal{Y}_u}[\log p(\theta|\mathcal{D}_0, \mathcal{D}_u, \mathcal{Y}_u)] \approx \mathbb{E}_{\mathcal{Y}'_u}[\log p(\theta|\mathcal{D}_0, \mathcal{D}'_u, \mathcal{Y}'_u)]$, where $\mathcal{Y}_u$ and $\mathcal{Y}'_u$ denote the predictive labels of $\mathcal{D}_u$ and $\mathcal{D}'_u$ respectively by the Bayesian discriminative model, that is, $p(\mathcal{Y}_u|\mathcal{D}_u, D_0) = \int_\theta p(\mathcal{Y}_u|\mathcal{D}_u,\theta)p(\theta|D_0)\mathrm{d}\theta$, and $p(\mathcal{Y}'_u|\mathcal{D}'_u, D_0) = \int_\theta p(\mathcal{Y}'_u|\mathcal{D}'_u,\theta)p(\theta|D_0)\mathrm{d}\theta$. Here $D'_u$ can be indicated by a core-set (Pinsler et al., 2019) that highly represents $\mathcal{D}_u$. Optimization tricks such as Frank-Wolfe optimization (Vavasis, 1992) then can be adopted to solve this problem.

**Motivations.** Eqs. (1) and (2) provide the Bayesian rules of BALD over model uncertainty and core-set construction respectively, which further attract the attention of the deep learning community. However, the two interpretations of BALD are limited by: 1) redundant information and 2) uninformative prior, where one major reason which causes these two issues is the poor initialization on the prior, i.e. $p(\mathcal{D}_0|\theta)$. For example, unbalanced label initialization on $\mathcal{D}_0$ usually leads to an uninformative prior, which further conducts the acquisitions of AL to select those unlabeled data from one or some fixed classes; highly-biased results with (Gao et al., 2020) redundant information are inevitable. Therefore, these two limitations affect each other.

## 4 GBALD

GBALD consists of two components: initial acquisitions based on core-set construction and model uncertainty estimation with those initial acquisitions.

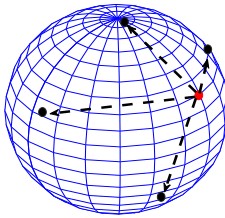
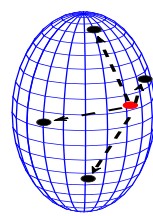

(a) Sphere geodesic     (b) Ellipsoid geodesic

### 4.1 GEOMETRIC INTERPRETATION OF CORE-SET

Modeling the complete data posterior over the parameter distribution can relieve the above two limitations of BALD. Typically, finding the acquisitions of AL is equivalent to approximating a core-set centered with spherical embeddings (Sener & Savarese, 2018). Let $w_i$ be the sampling weight of $x_i$, $\|w_i\|_0 \le N$, the core-set construction is to optimize:

Figure 2: Optimizing BALD with sphere and ellipsoid geodesics. Ellipsoid geodesic rescales the sphere geodesic to prevent the updates of the core-set towards the boundary regions of the sphere where the characteristics of the distribution cannot be captured. Black points denote the feasible updates of the red points. Dash lines denote the geodesics.

$$\min_{w} \left\| \underbrace{\sum_{i=1}^{N} \mathbb{E}_{y_i}\Big[\log p(y_i|x_i, \theta) + \mathrm{H}[y_i|x_i, \mathcal{D}_0]\Big]}_{\mathcal{L}} - \underbrace{\sum_{i=1}^{N} w_i \mathbb{E}_{y_i}\Big[\log p(y_i|x_i, \theta) + \mathrm{H}[y_i|x_i, \mathcal{D}_0]\Big]}_{\mathcal{L}(w)} \right\|^2, \quad (3)$$

where $\mathcal{L}$ and $\mathcal{L}(w)$ denote the full and expected (weighted) log-likelihoods, respectively (Campbell & Broderick, 2018; 2019). Specifically, $\sum_{i=1}^{N} \mathrm{H}[y_i|x_i, \mathcal{D}_0] = -\sum_{y_i} p(y_i|x_i, \mathcal{D}_0)\log(p(y_i|x_i, \mathcal{D}_0))$, where $p(y_i|x_i, \mathcal{D}_0) = \int_{\theta} p(y_i|x_i, \theta)p(\theta|\mathcal{D}_0)\mathrm{d}\theta$. Note $\|\cdot\|$ denotes the $\ell^2$ norm.

The approximation of Eq. (3) implicitly requires that the complete data log posterior of Eq. (2) w.r.t. $\mathcal{L}$ must be close to an expected posterior w.r.t. $\mathcal{L}(w)$ such that approximating a sparse subset for the original inputs by sphere geodesic search is feasible (see Figure 2(a)). Generally, solving this optimization is intractable due to cardinality constraint (Pinsler et al., 2019). Campbell & Broderick (2019) proposed to relax the constraint in Frank–Wolfe optimization, in which mapping $\mathcal{X}$ is usually performed in a Hilbert space (HS) with a bounded inner product operation. In this solution, the sphere embedded in the HS replaces the cardinality constraint with a polynomial constraint. However, the initialization on $\mathcal{D}_0$ affects the iterative approximation to $\mathcal{D}_u$ at the beginning of the geodesic search. Moreover, the posterior of $p(\theta|\mathcal{D}_0)$ is uninformative, if the initialized $\mathcal{D}_0$ is empty or not correct. Therefore, the typical Bayesian core-set construction of BALD cannot ideally fit an uninformative prior. The another geometric interpretation of core-set construction, such as $k$-centers (Sener & Savarese, 2018), is not restricted to this setting. We thus follow the construction of $k$-centers to find the core-set.

**$k$-centers.** Sener & Savarese (2018) proposed a core-set representation approach for active deep learning based on $k$-centers. This approach can be adopted in core-set construction of BALD without the help of the discriminative model. Therefore, the uninformative prior has no further influence on the core-set. Typically, the $k$-centers approach uses a greedy strategy to search the data $\widetilde{x}$ whose nearest distance to elements of $\mathcal{D}_0$ is the maximal:

$$\widetilde{x} = \arg\max_{x_i \in \mathcal{D}_u} \min_{c_i \in \mathcal{D}_0} \|x_i - c_i\|, \quad (4)$$

then $\mathcal{D}_0$ is updated by $\mathcal{D}_0 \cup \{\widetilde{x}, \widetilde{y}\}$, $\mathcal{D}_u$ is updated by $\mathcal{D}_u \backslash \widetilde{x}$, where $\widetilde{y}$ denotes the output of $\widetilde{x}$. This max-min operation usually performs $k$ times to construct the centers.

From the view of geometry, $k$-centers can be deemed as the core-set construction via spherical geodesic search (Bādoiu et al., 2002; Har-Peled & Mazumdar, 2004). Specifically, the max-min

optimization guides $\mathcal{D}_0$ to be updated into one data, which draws the longest line segment from $x_i, \forall i$ across the sphere center. The iterative update on $\widetilde{x}$ is then along its unique diameter through the sphere center. However, this greedy optimization has large probability that yields the core-set to fall into boundary regions of the sphere, which cannot capture the characteristics of the distribution.

## 4.2 INITIAL ACQUISITIONS BASED ON CORE-SET CONSTRUCTION

We present a novel greedy search which rescales the geodesic of a sphere into an ellipsoid following Eq. (4), in which the iterative update on the geodesic search is rescaled (see Figure 2(b)). We follow the importance sampling strategy to begin the search.

**Initial prior on geometry.** Initializing $p(\mathcal{D}_0|\theta)$ is performed with a group of internal spheres centered with $D_j, \forall j$, subjected to $D_j \in \mathcal{D}_0$, in which the geodesic between $\mathcal{D}_0$ and the unlabeled data is over those spheres. Since $\mathcal{D}_0$ is known, specification of $\theta$ plays the key role for initializing $p(\mathcal{D}_0|\theta)$. Given a radius $R_0$ for any observed internal sphere, $p(y_i|x_i, \theta)$ is firstly defined by

$$p(y_i|x_i, \theta) = \begin{cases} 1, & \exists j, \|x_i - D_j\| \le R_0, \\ \max\left\{ \dfrac{R_0}{\|x_i - D_j\|} \right\}, & \forall j, \|x_i - D_i\| > R_0, \end{cases} \tag{5}$$

thereby $\theta$ yields the parameter $R_0$. When the data is enclosed with a ball, the probability of Eq. (5) is 1. The data near the ball, is given a probability of $\max\left\{ \frac{R_0}{\|x_i - D_j\|} \right\}$ constrained by $\min\|x_i - D_j\|, \forall j$, i.e. the probability is assigned by the nearest ball to $x_i$, which is centered with $D_j$. From Eq. (3), the information entropy of $y_i \sim \{y_1, y_2, ..., y_N\}$ over $x_i \sim \{x_1, x_2, ..., x_N\}$ can be expressed as the integral regarding $p(y_i|x_i, \theta)$:

$$\sum_{i=1}^{N} \mathrm{H}(y_i|x_i, \mathcal{D}_0) = -\sum_{i=1}^{N} \int_\theta p(y_i|x_i, \theta)p(\theta|D_0)d\theta\log\left( \int_\theta p(y_i|x_i, \theta)p(\theta|D_0)\right)d\theta, \tag{6}$$

which can be approximated by $-\sum_{i=1}^{N} p(y_i|x_i, \theta)\log\left(p(y_i|x_i, \theta)\right)$ following the details of Eq. (3). In short, this indicates an approximation to the entropy over the entire outputs on $\mathcal{D}_u$ that assumes the prior $p(D_0|\theta)$ w.r.t. $p(y_i|x_i, \theta)$ is already known from Eq. (5).

**Max-min optimization.** Recalling the max-min optimization trick of $k$-centers in the core-set construction of (Sener & Savarese, 2018), the minimizer of Eq. (3) can be divided into two parts: $\min_{x^*} \mathcal{L}$ and $\max_w \mathcal{L}(w)$, where $\mathcal{D}_0$ is updated by acquiring $x^*$. However, updates of $\mathcal{D}_0$ decide the minimizer of $\mathcal{L}$ with regard to the internal spheres centered with $D_i, \forall i$. Therefore, minimizing $\mathcal{L}$ should be constrained by an unbiased full likelihood over $\mathcal{X}$ to alleviate the potential biases from the initialization of $\mathcal{D}_0$. Let $\mathcal{L}_0$ denote the unbiased full likelihood over $\mathcal{X}$ that particularly stipulates $\mathcal{D}_0$ as the $k$-means centers written as $\mathcal{U}$ of $\mathcal{X}$ which jointly draw the input distribution. We define $\mathcal{L}_0 = |\sum_{i=1}^{N} \mathbb{E}_{y_i}[\log p(y_i|x_i, \theta) + \mathrm{H}[y_i|x_i, \mathcal{U}]]|$ to regulate $\mathcal{L}$, that is

$$\min_{x^*} \|\mathcal{L}_0 - \mathcal{L}\|^2, \quad \text{s.t. } \mathcal{D}_0 = \mathcal{D}_0 \cup \{x^*, y^*\}, \mathcal{D}_u = \mathcal{D}_u \backslash x^*. \tag{7}$$

The other sub optimizer is $\max_w \mathcal{L}(w)$. We present a greedy strategy following Eq. (1):

$$\max_{1 \le i \le N} \min_{w_i} \sum_{i=1}^{N} w_i \mathbb{E}_{y_i}[\log p(y_i|x_i, \theta) + \mathrm{H}[y_i|x_i, \mathcal{D}_0]]$$
$$= \sum_{i=1}^{N} w_i \log p(y_i|x_i, \theta) - \sum_{i=1}^{N} w_i p(y_i|x_i, \theta)\log p(y_i|x_i, \theta), \tag{8}$$

which can be further written as: $\sum_{i=1}^{N} w_i \log p(y_i|x_i, \theta)(1 - \log p(y_i|x_i, \theta))$. Let $w_i = 1, \forall i$ for unbiased estimation of the likelihood $\mathcal{L}(w)$, Eq. (8) can be simplified as

$$\max_{x_i \in \mathcal{D}_u} \min_{D_j \in \mathcal{D}_0} \log p(y_i|x_i, \theta), \tag{9}$$

where $p(y_i|x_i, \theta)$ follows Eq. (5). Combining Eqs. (7) and (9), the optimization of Eq. (3) is then transformed as

$$x^* = \arg\max_{x_j \in \mathcal{D}_u} \min_{D_j \in \mathcal{D}_0} \left\{ \|\mathcal{L}_0 - \mathcal{L}\|^2 + \log p(y_j|x_j, \theta) \right\}, \tag{10}$$

where $\mathcal{D}_0$ is updated by acquiring $x^*$, i.e. $\mathcal{D}_0 = \mathcal{D}_0 \cup \{x^*, y^*\}$.

**Geodesic line.** For a metric geometry $M$, a geodesic line is a curve $\gamma$ which projects its interval $I$ to $M$: $I \to M$, maintaining everywhere locally a distance minimizer (Lou et al., 2020). Given a constant $\nu > 0$ such that for any $a, b \in I$ there exists a geodesic distance

$d(\gamma(a), \gamma(b)) \coloneqq \int_a^b \sqrt{g_{\gamma(t)}(\gamma'(t), \gamma'(t))} dt$, where $\gamma'(t)$ denotes the geodesic curvature, and $g$ denotes the metric tensor over $M$. Here, we define $\gamma'(t) = 0$, then $g_{\gamma(t)}(0, 0) = 1$ such that $d(\gamma(a), \gamma(b))$ can be generalized as a segment of a straight line: $d(\gamma(a), \gamma(b)) = \|a - b\|$.

**Ellipsoid geodesic distance.** For any observation points $p, q \in M$, if the spherical geodesic distance is defined as $d(\gamma(p), \gamma(q)) = \|p - q\|$. The affine projection obtains its ellipsoid interpretation: $d(\gamma(p), \gamma(q)) = \|\eta(p - q)\|$, where $\eta$ denotes the affine factor subjected to $0 < \eta < 1$.

**Optimizing with ellipsoid geodesic search.** The max-min optimization of Eq. (10) is performed on an ellipsoid geometry to prevent the updates of core-set towards the boundary regions, where ellipsoid geodesic line scales the original update on the sphere. Assume $x_i$ is the previous acquisition and $x^*$ is the next desired acquisition, the ellipsoid geodesic rescales the position of $x^*$ as $x_e^* = x_i + \eta(x^* - x_i)$. Then, we update this position of $x_e^*$ to its nearest neighbor $x_j$ in the unlabeled data pool, i.e. $\arg\min_{x_j \in \mathcal{D}_u} \|x_j - x_e^*\|$, also can be written as

$$\arg\min_{x_j \in \mathcal{D}_u} \left\| x_j - [x_i + \eta(x^* - x_i)] \right\|. \tag{11}$$

To study the advantage of ellipsoid geodesic search, Appendix B presents our generalization analysis.

### 4.3 MODEL UNCERTAINTY ESTIMATION WITH CORE-SET

GBALD starts the model uncertainty estimation with those initial core-set acquisitions, in which it introduces a ranking scheme to derive both informative and representative acquisitions.

**Single acquisition.** We follow (Gal et al., 2017) and use MC dropout to perform Bayesian inference on the model of the neural network. It then leads to ranking the informative acquisitions with batch sequences is with high efficiency. We first present the ranking criterion by rewriting Eq. (1) as batch returns:

$$\{x_1^*, x_2^*, ..., x_b^*\} = \arg\max_{\{\hat{x}_1, \hat{x}_2, ..., \hat{x}_b\} \subseteq \mathcal{D}_u} \mathrm{H}[\theta|\mathcal{D}_0] - \mathbb{E}_{\hat{y}_{1:b} \sim p(\hat{y}_{1:b}|\hat{x}_{1:b}, \mathcal{D}_0)} \Big[ \mathrm{H}[\theta|\hat{x}_{1:b}, \hat{y}_{1:b}, \mathcal{D}_0] \Big], \tag{12}$$

where $\hat{x}_{1:b} = \{\hat{x}_1, \hat{x}_2, ..., \hat{x}_b\}$, $\hat{y}_{1:b} = \{\hat{y}_1, \hat{y}_2, ..., \hat{y}_b\}$, $\hat{y}_i$ denotes the output of $\hat{x}_i$. The informative acquisition $x_t^*$ is then selected from the ranked batch acquisitions $\hat{x}_{1:b}$ due to the highest representation for the unlabeled data:

$$x_t^* = \arg\max_{x_i^* \in \{x_1^*, x_2^*, ..., x_b^*\}} \left\{ \max_{D_j \in \mathcal{D}_0} p(y_i|x_i^*, \theta) \coloneqq \frac{R_0}{\|x_i^* - D_j\|} \right\}, \tag{13}$$

where $t$ denotes the index of the final acquisition, subjected to $1 \leq t \leq b$. This also adopts the max-min optimization of $k$-centers in Eq. (4), i.e. $x_t^* = \arg\max_{x_i^* \in \{x_1^*, x_2^*, ..., x_b^*\}} \min_{D_j \in \mathcal{D}_0} \|x_i^* - D_j\|$.

**Batch acquisitions.** The greedy strategy of Eq. (13) can be written as a batch acquisitions by setting its output as a batch set, i.e.

$$\{x_{t_1}^*, ..., x_{t_{b'}}^*\} = \arg\max_{x_{t_1:t_{b'}}^* \subseteq \{x_1^*, x_2^*, ..., x_b^*\}} p(y_{t_1:t_{b'}}^*|x_{t_1:t_{b'}}^*, \theta), \tag{14}$$

where $x_{t_1:t_{b'}}^* = \{x_{t_1}^*, ..., x_{t_{b'}}^*\}$, $y_{t_1:t_{b'}}^* = \{y_{t_1}^*, ..., y_{t_{b'}}^*\}$, $y_{t_i}^*$ denotes the output of $x_{t_i}^*$, $1 \leq i \leq b'$, and $1 \leq b' \leq b$. This setting can be used to accelerate the acquisitions of AL in a large dataset. Appendix A presents the two-stage GBALD algorithm.

## 5 EXPERIMENTS

In experiments, we start by showing how BALD degenerates its performance with uninformative prior and redundant information, and show that how our proposed GBALD relieves theses limitations.

Our experiments discuss three questions: 1) is GBALD using core-set of Eq. (11) competitive with uninformative prior? 2) can GBALD using ranking of Eq. (14) improve informative acquisitions of model uncertainty? and 3) can GBALD outperform state-of-the-art acquisition approaches? Following the experiment settings of (Gal et al., 2017; Kirsch et al., 2019), we use MC dropout to implement the Bayesian approximation of DNNs. Three benchmark datasets are selected: MNIST, SVHN and CIFAR10. More experiments are presented in Appendix C.

### 5.1 UNINFORMATIVE PRIORS

As discussed in the introduction, BALD is sensitive to an uninformative prior, i.e. $p(\mathcal{D}_0|\theta)$. We thus initialize $\mathcal{D}_0$ from a fixed class of the tested dataset to observe its acquisition performance. Figure 3 presents the prediction accuracies of BALD with an acquisition budget of 130 over the training

set of MNIST, in which we randomly select 20 samples from the digit '0' and '1' to initialize $\mathcal{D}_0$, respectively. The classification model of AL follows a convolutional neural network with one block of [convolution, dropout, max-pooling, relu], with 32, 3x3 convolution filters, 5x5 max pooling, and 0.5 dropout rate. In the AL loops, we use 2,000 MC dropout samples from the unlabeled data pool to fit the training of the network following (Kirsch et al., 2019).

The results show BALD can slowly accelerate the training model due to biased initial acquisitions, which cannot uniformly cover all the label categories. Moreover, the uninformative prior guides BALD to unstable acquisition results. As the shown in Figure 3(b), BALD with Bathsize = 10 shows better performance than that of Batchsize =1; while BALD in Figure 3(a) keeps sta-

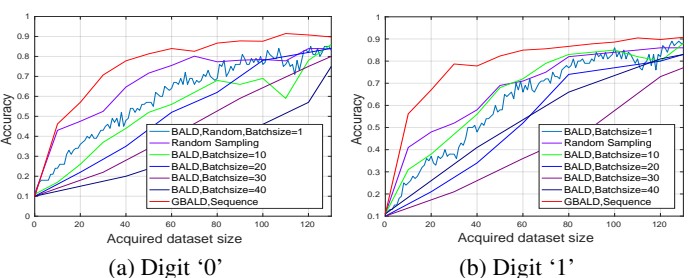

(a) Digit '0'  (b) Digit '1'

Figure 3: Acquisitions with uninformative priors from digit '0' and '1'.

ble performance. This is because the initial labeled data does not cover all classes and BALD with Batchsize =1 may further be misled to select those samples from one or a few fixed classes at the first acquisitions. However, Batchsize >1 may result in a random acquisition process that possibly covers more diverse labels at its first acquisitions. Another excursive result of BALD is that the increasing batch size cannot degenerate its acquisition performance in Figure 3(b). Specifically, Batchsize =10 > Batchsize =1 > Batchsize =20,40 > Batchsize =30, where '>' denotes 'better' performance; Batchsize = 20 achieves similar results of Batchsize =40. **This undermines the acquisition policy of BALD: its performance would degenerate when the batch size increases, and sometimes worse than random sampling. This also is the reason why we utilize a core-set to start BALD in our framework.**

Different to BALD, the core-set construction of GBALD using Eq. (11) provides a **complete label matching against all classes**. Therefore, it outperforms BALD with the batch sizes of 1, 10, 20, 30, and 40. As the shown learning curves in Figure 3, GBALD with a batch size of 1 and sequence size of 10 (i.e. breakpoints of acquired size are 10, 20, ..., 130) achieves significantly higher accuracies than BALD using different batch sizes since BALD misguides the network updating using poor prior.

## 5.2   IMPROVED INFORMATIVE ACQUISITIONS

Repeated or similar acquisitions delay the acceleration of the model training of BALD. Following the experiment settings of Section 5.1, we compare the best performance of BALD with a batch size of 1 and GBALD with different batch size parameters. Following Eq. (14), we set $b = \{3, 5, 7\}$ and $b'$=1, respectively, that means, we output the highest representative data from a batch of highly-informative acquisitions. Different settings on $b$ and $b'$ are used to observe the parameter perturbations of GBALD.

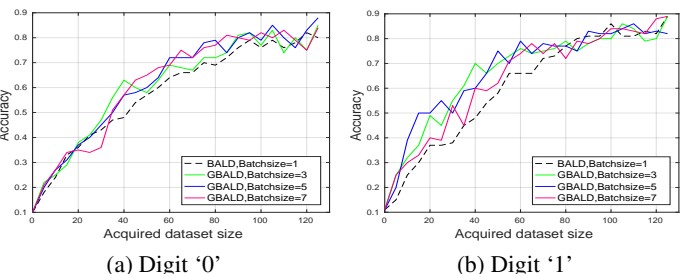

(a) Digit '0'  (b) Digit '1'

Figure 4: GBALD outperforms BALD using ranked informative acquisitions which cooperate with representation constraints.

Training by the same parameterized CNN model as Section 5.1, Figure 4 presents the acquisition performance of parameterized BALD and GBALD. As the learning curves shown, BALD cannot accelerate the model as fast as GBALD due to the repeated information over the acquisitions. For GBALD, it ranks the batch acquisitions of the highly-informative samples and selects the highest representative ones. By employing this special ranking strategy, GBALD can **reduce the probability of sampling nearby data of the previous acquisitions**. It is thus GBALD significantly outperforms BALD, even if we progressively increase the ranked batch size $b$.

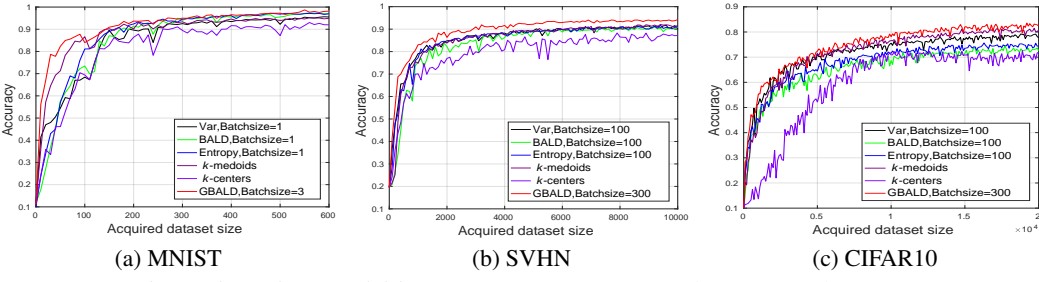

Figure 5: Active acquisitions on MNIST, SVHN, and CIFAR10 datasets.

Table 1: Mean±std of the test accuracies of the breakpoints of the learning curves on MNIST, SVHN, and CIFAR-10.

| Datasets | Algorithms | | | | | |
|---|---|---|---|---|---|---|
| | Var | BALD | Entropy | $k$-medoids | $k$-centers | GBALD |
| MNIST | 0.8419± 0.1721 | 0.8645±0.1909 | 0.8498±0.2098 | 0.8785±0.1433 | 0.8052±0.1838 | **0.9106±0.1296** |
| SVHN | 0.8535±0.1098 | 0.8510±0.1160 | 0.8294±0.1415 | 0.8498±0.1294 | 0.7909±0.1235 | **0.8885±0.1054** |
| CIFAR-10 | 0.7122±0.1034 | 0.6760±0.1023 | 0.6536±0.1038 | 0.71837±0.1245 | 0.5890±0.1758 | **0.7440±0.1087** |

## 5.3 ACTIVE ACQUISITIONS

GBALD using Eqs. (11) and (14) has been demonstrated to achieve successful improvements over BALD. We thus combine these two components into a uniform framework. Figure 5 reports the AL accuracies using different acquisition algorithms on the three image datasets. The selected baselines follow (Gal et al., 2017) including 1) maximizing the variation ratios (Var), 2) BALD, 3) maximizing the entropy (Entropy), 4) $k$-medoids, and one greedy 5) $k$-centers approach (Sener & Savarese, 2018). The network architecture is a three-layer MLP with three blocks of [convolution, dropout, max-pooling, relu], with 32, 64, and 128 3x3 convolution filters, 5x5 max pooling, and 0.5 dropout rate. In the AL loops, the MC dropout still randomly samples 2,000 data from the unlabeled data pool to approximate the training of the network architecture following (Kirsch et al., 2019). The initial labeled data of MNIST, SVHN and CIFAR-10 are 20, 1000, 1000 random samples from their full training sets. Details of baselines are presented in Appendix C.1.

The batch size of the compared baselines is 100, where GBALD ranks 300 acquisitions to select 100 data for the training, i.e. $b = 300, b' = 100$. As the learning curves shown in Figure 5, 1) $k$-centers algorithm performs more poorly than other compared baselines because the representative optimization with sphere geodesic usually falls into the selection of boundary data; 2) Var, Entropy and BALD algorithms cannot accelerate the network model rapidly due to highly-skewed acquisitions towards few fixed classes at its first acquisitions (start states); 3) $k$-medoids approach does not interact with the neural network model while directly imports the clustering centers into its training set; 4) The accuracies of the acquisitions of GBALD achieve better performance at the beginning than the Var, Entropy and BALD approaches which fed the training set of the network model via acquisition loops. In short, **the network is improved faster after drawing the distribution characteristics** of the input dataset with sufficient labels. GBALD thus consists of the representative and informative acquisitions in its uniform framework. Advantages of these two acquisition paradigms are integrated and present higher accuracies than any single paradigm.

Table 1 reports the mean±std values of the test accuracies of the breakpoints of the learning curves in Figure 5, where breakpoints of MNIST are $\{0, 10, 20, 30, ..., 600\}$, breakpoints of SVHN are $\{0, 100, 200, ..., 10000\}$, and breakpoints of CIFAR10 are $\{0, 100, 200, ..., 20000\}$. We then calculate their average accuracies and std values over these acquisition

Table 2: Number of acquisitions on MNIST, SVHN and CIFAR10 until 70%, 80%, and 90% accuracies are reached.

| Algorithms | Accuracies | | |
|---|---|---|---|
| | 70% | 80% | 90% |
| Var | 140/1,700/5,700 | 150/2,200/>20,000 | 210/>10,000/>6,100 |
| BALD | 110/1,700 /8,800 | 120 /2,300/>20,000 | 190/7,100 / >20,000 |
| Entropy | 110/1,900/11,200 | 150/2,400/>20,000 | 200/8,600/>20,000 |
| $k$-modoids | 70/1,700/5,900 | 90/2,200/16,000 | **170**/6,200 />20,000 |
| $k$-centers | 110/2,000/10,100 | 150/3,800/>20,000 | 280/>10,000/>20,000 |
| GBALD | **50/1,400/4,800** | **70/1,900/12,200** | **170/3,900**/>20,000 |

points. As the shown in Table 1, all std values around 0.1, yielding a norm value. Usually, an average

accuracy on a same acquisition size with different random seeds of DNNs, will result a small std value. Our mean accuracy spans across the whole learning curve.

The results show that 1) GBALD achieves the highest average accuracies; $k$-medoids is ranked the second amongst the compared baselines; 2) $k$-centers has ranked the worst accuracies amongst these approaches; 3) the others, which iteratively update the training model are ranked at the middle including BALD, Var and Entropy algorithms. Table 2 shows the acquisition numbers of achieving the accuracies of 70%, 80%, and 90% on the three datasets. The three numbers of each cell are the acquisition numbers over MNIST, SVHN, and CIFAR10, respectively. The results show that GBALD can use fewer acquisitions to achieve a desired accuracy than the other algorithms.

### 5.4  GBALD VS. BATCHBALD

Batch active deep learning was recently proposed to accelerate the training of a DNN model. In recent literature, BatchBALD (Kirsch et al., 2019) extended BALD with a batch acquisition setting to converge the network using fewer iteration loops. Different to BALD, BathBALD introduces diversity to avoid repeated or similar output acquisitions.

How to set the batch size of the acquisitions attracted our eyes before starting the experiments. It involves with whether our experiment settings are fair and reasonable. From a theoretical view, the larger the batch size, the worse the batch acquisitions will be. Experiments results of (Kirsch et al., 2019) also demonstrated this phenomenon. We thus set different batch sizes to run BatchBALD. Figure 6 reports the comparison results of BALD, BatchBALD, and our proposed GBALD following the experiment settings of Section 5.3. As the shown in this figure,

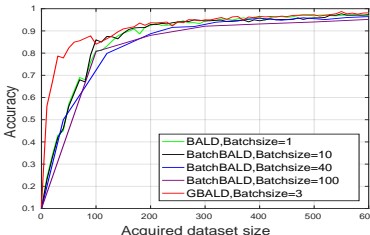

BatchBALD degenerates the test accuracies if we progressively increase the bath sizes, where BatchBALD with a batch size of 10 keeps similar learning curves as BALD. This means BatchBALD actually can accelerate BALD with a similar acquisition result if the batch size is not large. That means, if the batch size is between 2 to 10, BatchBALD will degenerate into BALD and maintains highly-consistent results.

Also because of this, BatchBALD has the same sensitivity to the uninformative prior. For our GBALD, the core-set solicits sufficient data which properly matches the input distribution (w.r.t. acquired data set size ≤ 100), providing powerful input features to start the DNN model (w.r.t.

Figure 6: Comparisons of BALD, Batch-BALD, and GBALD of active acquisitions on MNIST with bath settings.

acquired data set size > 100). Table 3 then presents the mean±std of breakpoints ($\{0, 10, 20, ..., 600\}$) of active acquisitions on MNIST with batch settings. The statistical results show GBALD has much higher mean accuracy than BatchBALD with different bath sizes. Therefore, evaluating the model uncertainty of DNN using highly-representative core-set samples can improve the performance of the neural network.

## 6  CONCLUSION

We have introduced a novel Bayesian AL framework, GBALD, from the geometric perspective, which seamlessly incorporates representative (core-set) and informative (model uncertainty estimation) acquisitions to accelerate the training of a DNN model. Our GBALD yields significant improvements over BALD, flexibly resolving the limitations of an uninformative prior and redundant information by optimizing the acquisition on an ellipsoid. Generalization analysis has asserted that training with ellipsoid has

Table 3: Mean±std of BALD, BatchBALD, and GBALD of active acquisitions on MNIST with batch settings.

| Algorithms | Batch sizes | Accuracies |
|---|---|---|
| BALD | 1 | 0.8654±0.0354 |
| BatchBALD | 10 | 0.8645±0.0365 |
| BatchBALD | 40 | 0.8273±0.0545 |
| BatchBALD | 100 | 0.7902±0.0951 |
| GBALD | 3 | 0.9106±0.1296 |

tighter lower error bound and higher probability to achieve a zero error than training with a typical sphere. Compared to the representative or informative acquisition algorithms, experiments show that our GBALD spends much fewer acquisitions to accelerate the accuracy. Moreover, it keeps slighter accuracy reduction than other baselines against repeated and noisy acquisitions.

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

## A. Two-stage GBALD Algorithm

The two-stage GBALD algorithm is described as follows: 1) construct core-set on ellipsoid (Lines 3 to 13), and 2) estimate model uncertainty with a deep learning model (Lines 14 to 21). Core-set construction is derived from the max-min optimization of Eq. (10), then updated with ellipsoid geodesic w.r.t. Eq. (11), where $\theta$ yields a geometric probability model w.r.t. Eq. (5). Importing the core-set into $\mathcal{D}_0$ derives the deep learning model to return $b$ informative acquisitions one time, where $\theta$ yields a deep learning model. Ranking those samples, we select $b'$ samples with the highest representations as the batch outputs. The iterations of batch acquisitions stop until its budget is exhaust. The final update on $\mathcal{D}_0$ is our acquisition set of AL.

**Details of the hyperparameter settings are presented at Appendix C.6.**

---

**Algorithm 1:** Two-stage GBALD Algorithm

---

1 **Input:** Data set $\mathcal{X}$, core-set size $N_{\mathcal{M}}$, batch returns $b$, batch output $b'$, iteration budget $\mathcal{A}$.
2 **Initialization:** $\alpha \leftarrow 0$, core-set $\mathcal{M} \leftarrow \varnothing$.
3 **Stage ① begins:**
4 Initialize $\theta$ to yield a geometric probability model w.r.t. Eq. (5).
5 Perform $k$-means to initialize $\mathcal{U}$ to $\mathcal{D}_0$.
6 Core-set construction begins by acquiring $x_i^*$,
7 **for** $i \leftarrow 1, 2, ..., N_{\mathcal{M}}$ **do**

8 $\quad$ $x_i^* \leftarrow \arg\max_{x_i \in \mathcal{D}_u} \ \min_{D_i \in \mathcal{D}_0} \left\{ \left\| \mathcal{L}_0 - \mathcal{L} \right\|^2 + \log p(y_i|x_i, \theta) \right\}$, where

$\quad$ $\mathcal{L}_0 \leftarrow \left| \sum_{i=1}^N \mathbb{E}_{y_i}[\log p(y_i|x_i, \theta) + \mathrm{H}[y_i|x_i, \mathcal{U}]] \right|$.

9 $\quad$ Ellipsoid geodesic line scales $x_i^*$: $x_i^* \leftarrow \arg\min_{x_j \in \mathcal{D}_u} \left\| x_j - [x_i + \eta(x^* - x_i)] \right\|$.
10 $\quad$ Update $x_i^*$ into core-set $\mathcal{M}$: $\mathcal{M} \leftarrow x_i^* \cup \mathcal{M}$.
11 $\quad$ Update $N \leftarrow N - 1$.
12 **end**
13 Import core-set to update $\mathcal{D}_0$: $\mathcal{D}_0 \leftarrow \mathcal{M} \cup \mathcal{U}'$, where $\mathcal{U}'$ updates each element of $\mathcal{U}$ into their nearest samples in $\mathcal{X}$.
14 **Stage ② begins:**
15 Initialize $\theta$ to yield a deep learning model.
16 **while** $\alpha < \mathcal{A}$ **do**
17 $\quad$ Return $b$ informative deep learning acquisitions in one budget:

$\quad$ $\{x_1^*, x_2^*, ..., x_b^*\} \leftarrow \arg\max_{x \in \mathcal{D}_u} \mathrm{H}[\theta|\mathcal{D}_0] - \mathbb{E}_{y \sim p(y|x, \mathcal{D}_0)}\Big[\mathrm{H}[\theta|x, y, \mathcal{D}_0]\Big]$.

18 $\quad$ Rank $b'$ informative acquisitions with the highest geometric representativeness:
$\quad$ $\{x_{t_1}^*, ..., x_{t_{b'}}^*\} \leftarrow \arg\max_{x_i^* \in \{x_1^*, x_2^*, ..., x_b^*\}} p(y_i|x_i^*, \theta)$.
19 $\quad$ Update $\{x_{t_1}^*, ..., x_{t_{b'}}^*\}$ into $\mathcal{D}_0$: $\mathcal{D}_0 \leftarrow \mathcal{D}_0 \cup \{x_{t_1}^*, ..., x_{t_{b'}}^*\}$.
20 $\quad$ $\alpha \leftarrow \alpha + 1$.
21 **end**
22 **Output:** final update on $\mathcal{D}_0$.

---

## B. Generalization errors of geodesic search with sphere and ellipsoid

Optimizing with ellipsoid geodesic linearly rescales the spherical search on tighter geometric object. The inherent motivation is that the ellipsoid can prevent the updates of core-set towards the boundary regions of the sphere. This section presents generalization error analysis from geometry, which provides feasible error guarantees for geodesic search with ellipsoid, following the perceptron analysis. Proofs are presented in Appendix D.

## B.1 Error bounds of geodesic search with sphere

Given a perceptron function $h := \mathbf{w_1 x_1} + \mathbf{w_2 x_2} + \mathbf{w_3}$, the classification task is over two classes $A$ and $B$ embedded in a three-dimensional geometry. Let $S_A$ and $S_B$ be the spheres that tightly cover $A$ and $B$, respectively, where $S_A$ is with a center $c_a$ and radius $R_a$, and $S_B$ is with a center $c_b$ and radius $R_b$. Under this setting, generalization analysis is presented.

**Theorem 1.** *Given a linear perceptron function $h = \mathbf{w_1 x_1} + \mathbf{w_2 x_2} + \mathbf{w_3}$ that classifies $A$ and $B$, and a sampling budget $k$, with representation sampling over $S_A$ and $S_B$, the minimum distances to the boundaries of $S_A$ and $S_B$ of that representation data are defined as $d_a$ and $d_b$, respectively. Let $err(h, k)$ be the classification error rate with respect to $h$ and $k$, $\frac{\pi}{\varphi} = \arcsin \frac{R_a - d_a}{R_a}$, we then have an inequality of error:*

$$\min\left\{\frac{4R_a^3 - (2R_a + t_k)(R_a - t_k)^2}{4R_a^3 + 4R_b^3}, \frac{4R_b^3 - (2R_b + t_k')(R_b - t_k')^2}{4R_b^3 + 4R_a^3}\right\} < err(h, k) < \frac{1}{k},$$

*where* $t_k = \frac{R_a^2}{3} + \sqrt[3]{-\frac{\mu_k}{2\pi} + \sqrt{\frac{\mu_k^2}{4\pi^2} - \frac{\pi^3 R_a^3}{27\pi^3}}} + \sqrt[3]{-\frac{\mu_k}{2\pi} - \sqrt{\frac{\mu_k^2}{4\pi^2} - \frac{\pi^3 R_a^3}{27\pi^3}}}$, $\mu_k = \left(\frac{2k-4}{3k} - \frac{1}{\varphi}\cos\frac{\pi}{\varphi}\right)\pi R_a^3 -$

$\frac{4\pi R_b^3}{3k}$, $t_k' = \frac{R_b^2}{3} + \sqrt[3]{-\frac{\mu_k'}{2\pi} + \sqrt{\frac{\mu_k'^2}{4\pi^2} - \frac{\pi^3 R_b^3}{27\pi^3}}} + \sqrt[3]{-\frac{\mu_k'}{2\pi} - \sqrt{\frac{\mu_k'^2}{4\pi^2} - \frac{\pi^3 R_b^3}{27\pi^3}}}$, *and* $\mu_k' = \left(\frac{2k-4}{3k} - \frac{1}{\varphi}\cos\frac{\pi}{\varphi}\right)\pi R_b^3 - \frac{4\pi R_a^3}{3k}$.

## B.2 Error bounds of geodesic search with ellipsoid

Given class $A$ and $B$ are tightly covered by ellipsoid $E_a$ and $E_b$ in a three-dimensional geometry. Let $R_{a_1}$ be the polar radius of $E_a$, $\{R_{a_2}, R_{a_3}\}$ be the equatorial radii of $E_a$, $R_{b_1}$ be the polar radius of $E_b$, and $\{R_{b_2}, R_{b_3}\}$ be the equatorial radii of $E_b$, the generalization analysis is ready to present following these settings.

**Theorem 2.** *Given a linear perceptron function $h = \mathbf{w_1 x_1} + \mathbf{w_2 x_2} + \mathbf{w_3}$ that classifies $A$ and $B$, and a sampling budget $k$, with representation sampling over $E_a$ and $E_b$, the minimum distance to the boundaries of $E_a$ and $E_b$ of that representation data are defined as $d_a$ and $d_b$, respectively. Let $err(h, k)$ be the classification error rate with respect to $h$ and $k$, $\frac{\pi}{\varphi} = \arcsin \frac{R_a - d_a}{R_a}$, we then have an inequality of error:*

$$\min\left\{\frac{4\prod_i R_{a_i} - (2R_{a_1} + \lambda_k)(R_{a_1} - \lambda_k)^2}{4\prod_i R_{a_i} + 4\prod_i R_{b_i}}, \frac{4\prod_i R_{b_i} - (2R_{b_1} + \lambda_k')(R_{b_1} - \lambda_k')^2}{4\prod_i R_{b_i} + 4\prod_i R_{a_i}}\right\} < err(h, k) < \frac{1}{k},$$

*where* $i = 1, 2, 3$, $\lambda_k = \frac{R_{a_1}^2}{3} + \sqrt[3]{-\frac{\sigma_k}{2\pi} + \sqrt{\frac{\sigma_k^2}{4\pi^2} - \frac{\pi^3 R_{a_1}^3}{27\pi^3}}} + \sqrt[3]{-\frac{\sigma_k}{2\pi} - \sqrt{\frac{\sigma_k^2}{4\pi^2} - \frac{\pi^3 R_{a_1}^3}{27\pi^3}}}$, $\sigma_k = \left(\frac{2k-4}{3k} - \right.$

$\left.\frac{\pi R_{a_1}}{2\varphi}\right)\pi \prod_i R_{a_i} - \frac{4\pi \prod_i R_{b_i}}{3k}$, $\lambda_k' = \frac{R_{b_1}^2}{3} + \sqrt[3]{-\frac{\sigma_k}{2\pi} + \sqrt{\frac{\sigma_k'^2}{4\pi^2} - \frac{\pi^3 R_{b_1}^3}{27\pi^3}}} + \sqrt[3]{-\frac{\sigma_k}{2\pi} - \sqrt{\frac{\sigma_k^2}{4\pi^2} - \frac{\pi^3 R_{b_1}^3}{27\pi^3}}}$, *and* $\sigma_k' = \left(\frac{2k-4}{3k} - \frac{\pi R_{b_1}}{2\varphi}\right)\pi \prod_i R_{b_i} - \frac{4\pi \prod_i R_{a_i}}{3k}$.

## B.3 Probability bounds of achieving a zero generalization error

Let $\Pr[err(h, k) = 0]_{\text{Sphere}}$ and $\Pr[err(h, k) = 0]_{\text{Ellipsoid}}$ be the probabilities of achieving a zero generalization error of geodesic search with sphere and ellipsoid, respectively, we present their inequality relationship.

**Theorem 3.** *Based on $\gamma$-tube theory (Ben-David & Von Luxburg, 2008) of clustering stability, the probability of achieving a zero generalization error of geodesic search with ellipsoid can be defined as the volume ratio of $\frac{\text{Vol(Tube)}}{\text{Vol(Sphere)}}$. Then, we have:*

$$\Pr[err(h, k) = 0]_{\text{Sphere}} = 1 - \frac{t_k^3}{R_a^3},$$

*where $t_k$ keeps consistent with Theorem 1.*

**Theorem 4.** *Based on $\gamma$-tube theory (Ben-David & Von Luxburg, 2008) of clustering stability, the probability of achieving a zero generalization error of geodesic search with ellipsoid can be defined as the volume ratio of $\frac{\text{Vol(Tube)}}{\text{Vol(Ellipsoid)}}$. Then, we have:*

$$\Pr[err(h,k)=0]_{\text{Ellipsoid}} = 1 - \frac{\lambda_{k_1}\lambda_{k_2}\lambda_{k_3}}{R_{a_1}R_{a_2}R_{a_3}},$$

*where $\lambda_{k_i} = \frac{R_{a_i}^2}{3} + \sqrt[3]{-\frac{\sigma_{k_i}}{2\pi} + \sqrt{\frac{\sigma_{k_i}^2}{4\pi^2} - \frac{\pi^3 R_{a_i}^3}{27\pi^3}}} + \sqrt[3]{-\frac{\sigma_{k_i}}{2\pi} - \sqrt{\frac{\sigma_{k_i}^2}{4\pi^2} - \frac{\pi^3 R_{a_i}^3}{27\pi^3}}}$, and $\sigma_{k_i} = (\frac{2k-4}{3k} - \frac{\pi R_{a_i}}{2\varphi})\pi R_{a_1} R_{a_2} R_{a_3} - \frac{4\pi R_{b_1} R_{b_2} R_{b_3}}{3k}$, $i = 1, 2, 3$.*

### B.4 MAIN THEORETICAL RESULTS OF GEODESIC SEARCH WITH ELLIPSOID

#### TIGHTER LOWER ERROR BOUND

Let $\text{lower}[err(h,k)]_{\text{Sphere}}$ and $\text{lower}[err(h,k)]_{\text{Ellipsoid}}$ be the lower bounds of the generalization errors of geodesic search with sphere and ellipsoid, respectively. With $R_{a_1} < R_a$, compare Theorem 1 and Theorem 2, we have the following proposition.

**Proposition 1.** *Given a linear perceptron function $h = \mathbf{w_1 x_1} + \mathbf{w_2 x_2} + \mathbf{w_3}$ that classifies $A$ and $B$, and a sampling budget $k$. With representation sampling over $S_a$ and $S_b$, let $err(h,k)$ be the classification error rate with respect to $h$ and $k$, the lower bounds of geodesic search with sphere and ellipsoid satisfy:* $\text{lower}[err(h,k)]_{\text{Ellipsoid}} < \text{lower}[err(h,k)]_{\text{Sphere}}$.

#### HIGHER PROBABILITY OF ACHIEVING A ZERO ERROR

Let $\Pr[err(h,k)=0]_{\text{Sphere}}$ and $\Pr[err(h,k)=0]_{\text{Ellipsoid}}$ be the probabilities of achieving a zero generalization error of geodesic search with sphere and ellipsoid, respectively. Their relationship is presented in Proposition 2.

**Proposition 2.** *Given a linear perceptron function $h = \mathbf{w_1 x_1} + \mathbf{w_2 x_2} + \mathbf{w_3}$ that classifies $A$ and $B$, and a sampling budget $k$. With representation sampling over $E_a$ and $E_b$, let $err(h,k)$ be the classification error rate with respect to $h$ and $k$, the probabilities of geodesic search with sphere and ellipsoid satisfy:* $\Pr[err(h,k)=0]_{\text{Ellipsoid}} > \Pr[err(h,k)=0]_{\text{Sphere}}$.

With these theoretical results, geometric interpretation of geodesic search over ellipsoid is more effective than sphere due to tighter lower error bound and higher probability to achieve zero error. Theorems 6 and 7 of Appendix D then present a high-dimensional generalization for the above theoretical results in terms of the volume functions of sphere and ellipsoid.

## C. EXPERIMENTS

### C.1 BASELINES

To evaluate the performance of GBALD, several typical baselines from the latest deep AL literatures are selected.

- Bayesian active learning by disagreement (BALD) (Houlsby et al., 2011). It has been introduced in Section 3.
- Maximize Variation Ratio (Var) (Gal et al., 2017). The algorithm chooses the unlabeled data that maximizes its variation ratio of the probability:

$$x^* = \arg\max_{x \in \mathcal{D}_u} \left\{ 1 - \max_{y \in \mathcal{Y}} \Pr(y|, x, \mathcal{D}_0)) \right\}. \tag{15}$$

- Maximize Entropy (Entropy) (Gal et al., 2017). The algorithm chooses the unlabeled data that maximizes the predictive entropy:

$$x^* = \arg\max_{x \in \mathcal{D}_u} \left\{ -\sum_{y \in \mathcal{Y}} \Pr(y|x, \mathcal{D}_0)) \log\left(\Pr(y|x, \mathcal{D}_0)\right) \right\}. \tag{16}$$

Table 4: Mean±std of active acquisitions on SVHN with 5,000 and 10,000 repeated samples.

| Algorithms | Accuracies | | |
|---|---|---|---|
| | 0 repeated samples | 5,000 repeated samples | 10,000 repeated samples |
| Var | 0.8535±0.1098 | 0.8478±0.1074 | 0.8281±0.1082 |
| BALD | 0.8510±0.1160 | 0.8119±0.1216 | 0.7689±0.1288 |
| GBALD | **0.8885±0.1054** | **0.8694±0.1032** | **0.8630±0.1002** |

- $k$-modoids (Park & Jun, 2009). A classical unsupervised algorithm that represents the input distribution with $k$ clustering centers:

$$\{x_1^*, x_2^*, ..., x_k^*\} = \underset{z_1, z_2, ..., z_k}{\arg\min} \Big\{ \sum_{i=1}^{k} \sum_{z_i \in \mathcal{X}^k} \|x_i - z_i\| \Big\}, \tag{17}$$

  where $\mathcal{X}^k$ denotes the $k$-th subcluster centered with $z_i$, and $z_i \in \mathcal{X}, \forall i..$

- Greedy $k$-centers ($k$-centers) (Sener & Savarese, 2018). A geometric core-set interpretation on sphere. See Eq. (4).

- BatchBALD (Kirsch et al., 2019). A batch extension of BALD which incorporates the diversity, not maximal entropy as BALD, to rank the acquisitions:

$$\{x_{t_1}^*, ..., x_{t_b}^*\} = \underset{x_{t_1}, ..., x_{t_b}}{\arg\max} \, \mathrm{H}(y_{t_1}, ...., y_{t_b}) - \mathrm{E}_{p(\theta|\mathcal{D}_0)}[\mathrm{H}(y_{t_1}, ....y_{t_b}|\theta)], \tag{18}$$

  where $\mathrm{H}(y_{t_1}, ...., y_{t_b})$ denote the expected entropy over all possible labels from $y_{t_1}$ to $y_{t_b}$ such that $\mathrm{H}(y_{t_1}, ...., y_{t_b}) = \mathrm{E}_p(y_{t_1}, ...y_{t_b})[-\mathrm{log}p(y_{t_1}, ...y_{t_b})]$, and $\mathrm{E}_{p(\theta|\mathcal{D}_0)}[\mathrm{H}(y_{t_1}, ....y_{t_b}|\theta)]$ is estimated by MC sampling (Roy & McCallum, 2001) (Osborne et al., 2012) a subset from $\mathcal{X}$ which approximates the parameter distributions of $\theta$.

The parameter settings of Eq. (5) are $R_0 = 2.0e + 3$ and $\eta = 0.9$. Accuracy of each acquired dataset size of the experiments are averaged over 3 runs.

## C.2 ACTIVE ACQUISITIONS WITH REPEATED SAMPLES

Repeatedly collecting samples in the establishment of a database is very common. Those repeated samples may be continuously evaluated as the primary acquisitions of AL due to the lack of one or more kinds of class labels. Meanwhile, this situation may lead the evaluation of the model uncertainty to fall into repeated acquisitions. To respond this collecting situation, we compare the acquisition performance of BALD, Var, and GBALD using 5,000 and 10,000 repeated samples from the first 5,000 and 10,000 unlabeled data of SVHN, respectively. In addition, the unsupervised algorithms which do not interact with the network architecture, such as $k$-medoids and $k$-centers, have been shown that they cannot accelerate the training in terms of the experiment results of Section 5.3. Thus, we are no longer studying their performance. The network architecture still follows the settings of the MLP as Section 5.3.

The acquisition results over the repeated SVHN datasets are presented in Figure 7. The batch sizes of the compared baselines are 100, where GBALD ranks 300 acquisitions to select 100 data for the training, i.e. $b = 300, b' = 100$. The mean±std values of these baselines of the breakpoints (i.e. $\{0, 100, 200, ..., 10000\}$) are reported in Table 4. Results demonstrate that GBALD shows slighter perturbations on repeated samples than Var and BALD because it draws the core-set from the input distribution as the initial acquisition, leading small probability to sample from one or more fixed class. In GBALD, the informative acquisitions constrained with geometric representations further scatter the acquisitions spread in different classes. However, Var and BALD algorithms have no particular schemes against the repeated acquisitions. The maximizer on the model uncertainty may be repeatedly produced by those repeated samples. In additional, the unsupervised algorithms such as $k$-medoids and $k$-centers don not have these limitations, but cannot accelerate the training since there has no interactions with the network architecture.

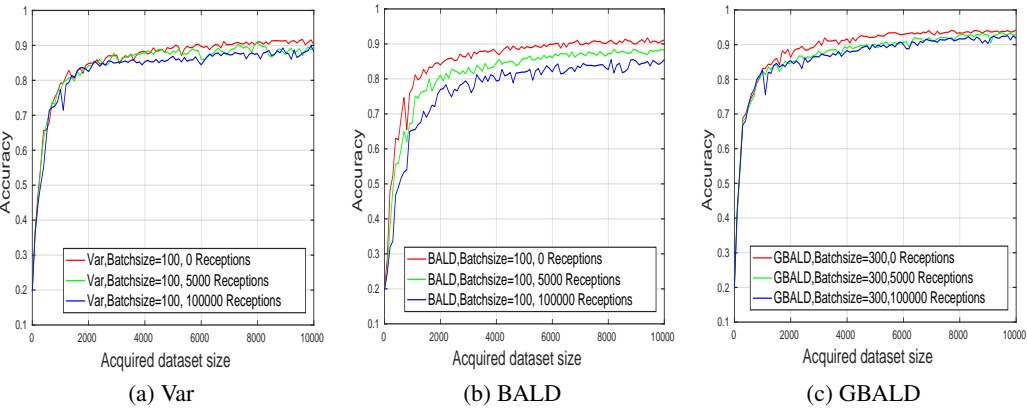

Figure 7: Active acquisitions on SVHN with 5,000 and 10,000 repeated samples.

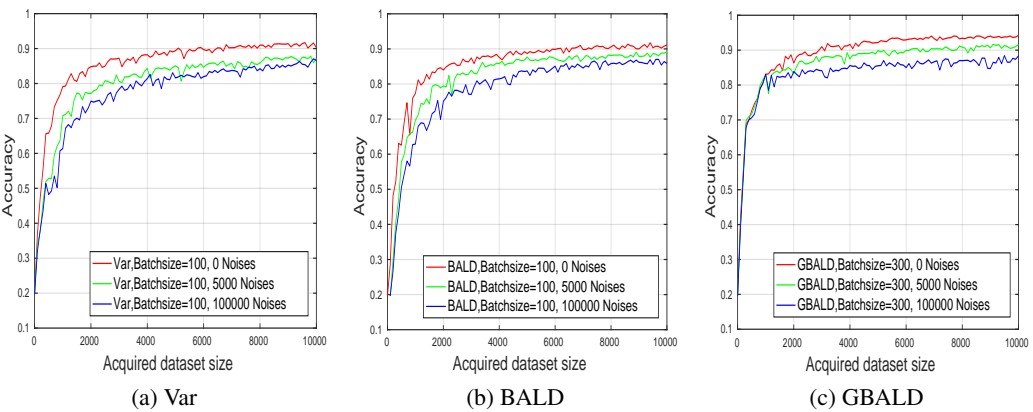

Figure 8: Active noisy acquisitions on SVHN with 5,000 and 10,000 noisy labels.

### C.3 ACTIVE ACQUISITIONS WITH NOISY SAMPLES

Noisy labels (Golovin et al., 2010) (Han et al., 2018) are inevitable due to human errors in data annotation. Training on noisy labels, the neural network model will degenerate its inherent properties. To assess the perturbations of the above acquisition algorithms against noisy labels, we organize the following experiment scenarios: we select the first 5,000 and 10,000 samples respectively from the unlabeled data pool of the MNIST dataset and reset their labels by shifting {'0','1',...,'8'} to {'1','2',...,'9'}, respectively. The network architecture follows MLP of Section 5.3. The selected baselines are Var and BALD.

Figure 7 presents the acquisition results of those baseline with noisy labels. The batch sizes of the compared baselines are 100, where GBALD ranks 300 acquisitions to select 100 data for the training, i.e. $b = 300, b' = 100$. Table 5 presents the mean±std values of the breakpoints (i.e. $\{0, 100, 200, ..., 10000\}$) over learning curves of Figure 8. The results further show that GBALD has smaller noisy perturbations than other baselines. For Var and BALD, model uncertainty leads high probabilities to sample those noisy data due to their greatly updating on the model.

### C.5 ACCELERATION OF ACCURACY

Accelerations of accuracy i.e. the first-orders of breakpoints of the learning curve, describe the efficiency of the active acquisition loops. Different to the accuracy curves, the acceleration curve reflects how active acquisitions help the convergence of the interacting DNN model.

Table 5: Mean±std of active noisy acquisitions on SVHN with 5,000 and 10,000 noises.

| Algorithms | Accuracies | | |
|---|---|---|---|
| | 0 noises | 5,000 noises | 10,000 noises |
| Var | 0.8535±0.1098 | 0.7980±0.1203 | 0.7702±0.1238 |
| BALD | 0.8510±0.1160 | 0.8205±0.1185 | 0.7849±0.1239 |
| GBALD | **0.8885±0.1054** | **0.8622±0.0991** | **0.8301±0.0916** |

We thus firstly present the acceleration curves of different baselines on MNIST, SVHN, and CIFAR10 datasets following the experiments of Section 5.3. The acceleration curves of active acquisitions are drawn in Figure 9. Observing those acceleration curves of different algorithms clearly finds that, GBALD always keeps **higher accelerations of accuracy** than the other baselines against the three benchmark datasets. This revels the reason of why GBALD can derive more informative and representative data to maximally update the DNN model.

The acceleration curves of active acquisitions with repeated samples of Appendix C.2 are presented in Figure 10. As the shown in this figure, GBALD presents **slighter perturbations** to the number of repeated samples than that of Var and BALD due to its effective ranking scheme on optimizing model uncertainty of DNN. The acceleration curves of active noisy acquisitions of Section Appendix C.3 are drawn in Figure 11. Compared to Figure 7, it presents more intuitive descriptions for the noisy perturbations to different baselines. With horizontal comparisons to acceleration curves of Var and BALD, our proposed GBALD has smaller noisy perturbations due to 1) the powerful core-set which properly captures the input distribution, and 2) highly representative and informative acquisitions of model uncertainty.

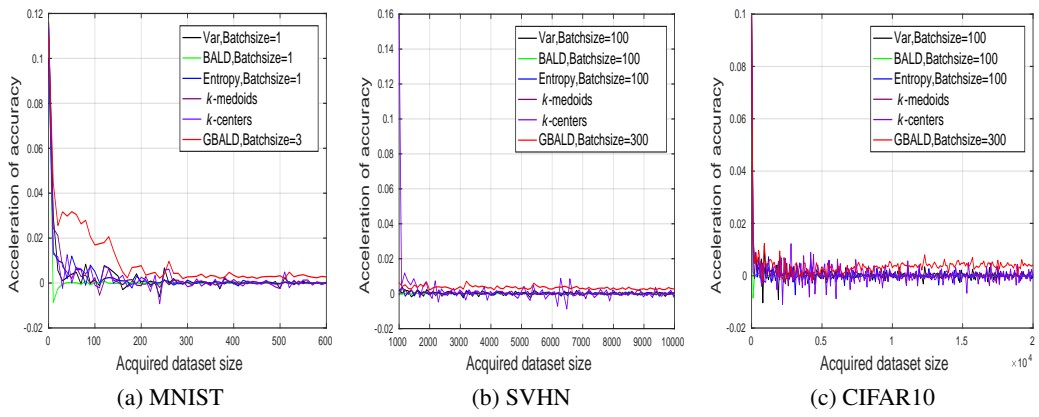

Figure 9: Accelerations of accuracy of different baselines on MNIST, SVHN, and CIFAR10 datasets.

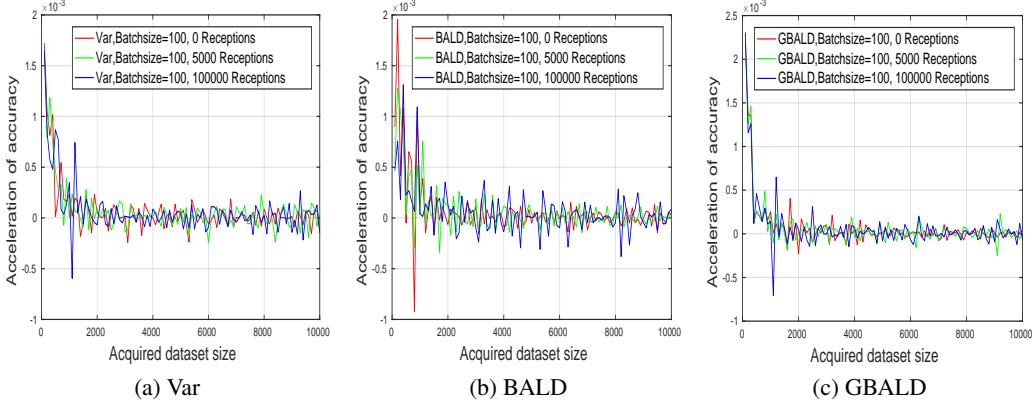

Figure 10: Accelerations of accuracy of active acquisitions on SVHN with 5,000 and 10,000 repeated samples.

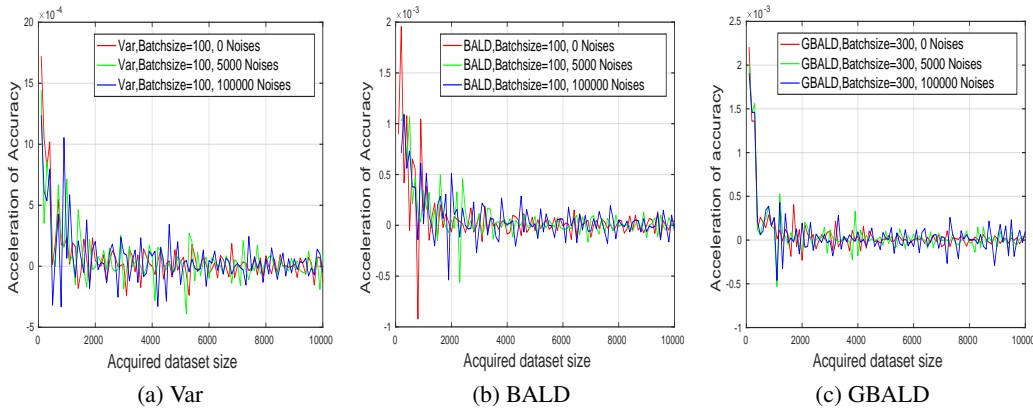

(a) Var       (b) BALD       (c) GBALD

Figure 11: Accelerations of accuracy of active noisy acquisitions on SVHN with 5,000 and 10,000 noisy labels.

Table 6: Relationship of accuracies and sizes of core-set on SVHN.

| Size of core-set | Accuracies | | |
|---|---|---|---|
| | Start accuracy | Ultimate accuracy | Mean±std accuracy |
| $N_{\mathcal{M}}$ = 1,000 | 0.8790 | 0.9344 | 0.9134±0.0169 |
| $N_{\mathcal{M}}$ = 2,000 | 0.8898 | 0.9212 | 0.9151±0.0148 |
| $N_{\mathcal{M}}$ = 3,000 | 0.8848 | **0.9364** | 0.9173±0.0138 |
| $N_{\mathcal{M}}$ = 4,000 | 0.8811 | 0.9271 | 0.9146±0.0165 |
| $N_{\mathcal{M}}$ = 5,000 | **0.8959** | 0.9342 | **0.9197±0.0117** |

C.6 HYPERPARAMETER SETTINGS

What is the proper time to start active acquisitions using Eq. (14) in GBALD framework? Does the ratio of core-set and model uncertainty acquisitions affect the performance of GBALD?

**We discuss the key hyperparameter of GBALD here: core-set size** $N_{\mathcal{M}}$**.** Table 6 presents the relationship of accuracies and the sizes of core-set, where the start accuracy denotes the test accuracy over the initial core-set, and the ultimate accuracy denotes the test accuracy over up to $Q = 20,000$ training data. Let $b = 1000, b' = 500$ in GBALD, $\mathcal{N}_{\mathcal{M}}$ be the number of the core-set size, the iteration budget $\mathcal{A}$ of GBALD then can be defined as $\mathcal{A} = (Q - \mathcal{N}_{\mathcal{M}})/b'$. For example, if the number of the initial core-set labels are set as $\mathcal{N}_{\mathcal{M}} = 1,000$, we have $\mathcal{A} = (Q - \mathcal{N}_{\mathcal{M}})/b' \approx 38$; if $\mathcal{N}_{\mathcal{M}} = 2,000$, then $\mathcal{A} = (Q - \mathcal{N}_{\mathcal{M}})/b' \approx 36$.

From Table 6, GBALD algorithm keep stable accuracies over the start, ultimate, and mean±std accuracies when there inputs more than 1,000 core-set labels. Therefore, drawing sufficient core-set labels using Eq. (10) to start the model uncertainty of Eq. (14) can maximize the performance of our GBALD framework.

**Hyperparameter settings on batch returns** $b$ **and bath outputs** $b'$**.** Experiments of Sections 5.1 and 5.2 used different $b$ and $b'$ to observe the parameter perturbations. No matter what the settings of $b'$ and $b$ are, GBALD still outperforms BALD. For single acquisition of GBALD, we suggest $b = 3$ and $b' = 1$. For bath acquisitions, the settings on $b'$ and $b$ are user-defined according the time cost and hardware resources.

**Hyperparameter setting on iteration budget** $\mathcal{A}$**.** Given the acquisition budget $Q$, let $b'$ be the number of the output returns at each loop, $\mathcal{N}_{\mathcal{M}}$ be the number of the core-set size, the iteration budget $\mathcal{A}$ of GBLAD then can be defined as $\mathcal{A} = (Q - \mathcal{N}_{\mathcal{M}})/b'$.

**Other hyperparameter settings.** Eq. (5) has one parameter $R_0$ which describes the geometry prior from probability. The default radius of the intern balls $R_0$ is used to legalize the prior and has no further influences on Eq. (10). It is set as $R_0 = 2.0e + 3$ for those three image datasets. Ellipsoid

geodesic is adjusted by $\eta$ which controls how far of the updates of core-set to the boundaries of the distributions. It is set as $\eta = 0.9$ in this paper.

### C.7 TWO-SIDED $t$-TEST

We present two-sided (two-tailed) $t$-test for the learning curves of Figure 5. Different to the mean± std of Table 1, $t$-test can enlarge the significant difference of those baselines. In typical $t$-test, the two groups of observations usually require a degree of freedom smaller than 30. However, the numbers of breakpoints of MNIST, SVHN, and CIFAR10 are 61, 101, and 201, respectively, thereby holding a degree of freedom of 60, 100, 200, respectively. It is thus we introduce $t$-test score to directly compare the significant difference of pairwise baselines.

$t$-test score between any pair group of breakpoints are defined as follows. Let $B_1 = \{\alpha_1, \alpha_2, ..., \alpha_n\}$ and $B_2 = \{\beta_1, \beta_2, ..., \beta_n\}$, there exists $t$-score of

$$t - \text{score} = \sqrt{n}\frac{\mu}{\sigma},$$

where $\mu = \frac{1}{n}\sum_{i=1}^{n}(\alpha_i - \beta_i)$, and $\sigma = \sqrt{\frac{1}{n-1}\sum_{i=1}^{n}(\alpha_i - \beta_i - \mu)^2}$.

In two-sided $t$-test, $B_1$ beats $B_2$ on breakpoints $\alpha_i$ and $\beta_i$ satisfying a condition of $t - \text{score} > \nu$; $B_2$ beats $B_1$ on breakpoints $\alpha_i$ and $\beta_i$ satisfying a condition of $t - \text{score} < -\nu$, where $\nu$ denotes the hypothesized criterion with a given confidence risk. Following (Ash et al., 2019), we add a penalty of $\frac{1}{e}$ to each pair of breakpoints, which further enlarges their differences in the aggregated penalty matrix, where $e$ denotes the number of $B_1$ beats $B_2$ on all breakpoints. All penalty values finally calculate their $L_1$ expressions.

Figure 12 presents the penalty matrix over learning curves of Figure 5. Column-wise values at the bottom of each matrix show the overall performance of the compared baselines. As the shown results, GBALD has significant performances than that of the other baselines over the three datasets. Especially for SVHN, it has superior performance.

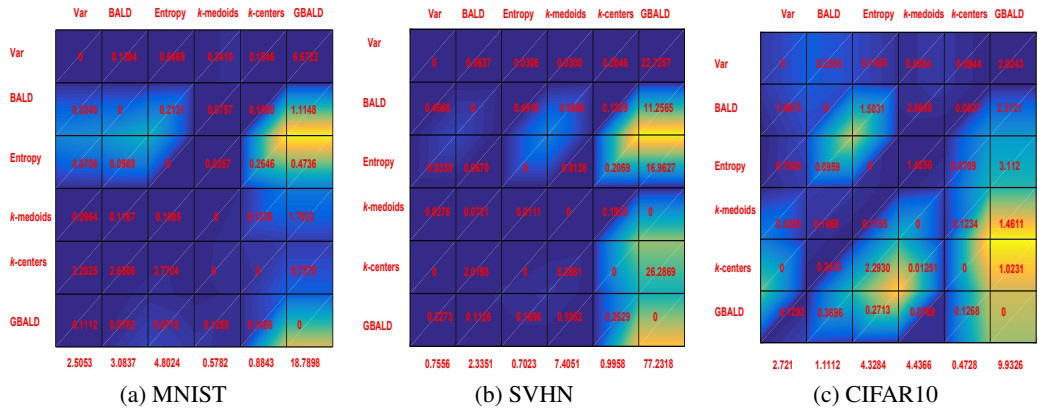

Figure 12: A pairwise penalty matrix over active acquisitions on MNIST, SVHN, and CIFAR10. Column-wise values at the bottom of each matrix show the overall performance of the compared baselines (larger value has more significant superior performance).

### D. PROOFS

We firstly present the generalization errors of $k = 3$ of AL with sphere as a case study. The assumption of the generalization analysis is described in Figure 13.

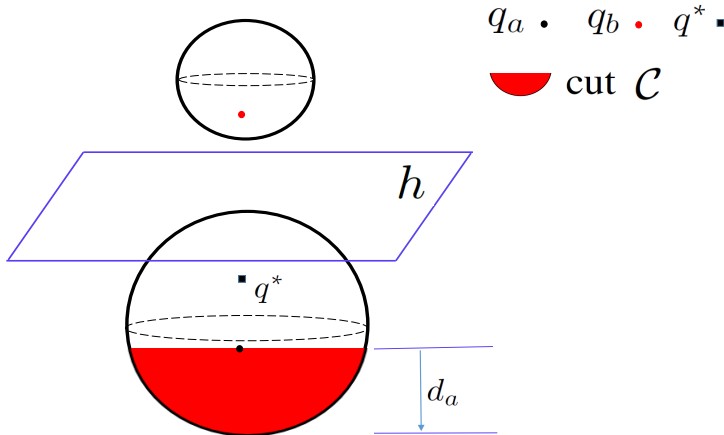

Figure 13: Assumption of the generalization analysis. The ball above $h$ denotes $S_b$, the ball below $h$ denotes $S_a$, and $R_b < R_a$. Spherical cap[1] of the half-sphere of $S_a$ is denoted as $\mathcal{D}$.

### D.1 CASE STUDY OF GENERALIZATION ANALYSIS OF $err(h,3)$ OF GEODESIC SEARCH WITH SPHERE

**Theorem 5.** *Given a linear perceptron function $h = \mathbf{w_1 x_1} + \mathbf{w_2 x_2} + \mathbf{w_3}$ that classifies $A$ and $B$, and a sampling budget $k$. With representation sampling over $S_a$ and $S_b$, the minimum distance to the boundaries of $S_A$ and $S_B$ of that representation data are defined as $d_a$ and $d_b$, respectively. Let $err(h,k)$ be the classification error rate with respect to $h$ and $k$, $\frac{\pi}{\varphi} = \arcsin\frac{R_a - d_a}{R_a}$, we have an inequality of error:*

$$\min\left\{\frac{(2R_a + t)(R_a - t)^2}{4R_a^3 + 4R_b^3}, \frac{(2R_b + t')(R_b - t')^2}{4R_b^3 + 4R_a^3}\right\} < err(h,3) < 0.3334,$$

*where $t = \frac{R_a^2}{3} + \sqrt[3]{-\frac{\mu}{2\pi} + \sqrt{\frac{\mu^2}{4\pi^2} - \frac{\pi^3 R_a^3}{27\pi^3}}} + \sqrt[3]{-\frac{\mu}{2\pi} - \sqrt{\frac{\mu^2}{4\pi^2} - \frac{\pi^3 R_a^3}{27\pi^3}}}$, $\mu = (\frac{2}{9} - \frac{1}{\varphi}cos\frac{\pi}{\varphi})\pi R_a^3 - \frac{4\pi R_b^3}{9}$, $t' = \frac{R_b^2}{3} + \sqrt[3]{-\frac{\mu'}{2\pi} + \sqrt{\frac{\mu'^2}{4\pi^2} - \frac{\pi^3 R_b^3}{27\pi^3}}} + \sqrt[3]{-\frac{\mu'}{2\pi} - \sqrt{\frac{\mu'^2}{4\pi^2} - \frac{\pi^3 R_b^3}{27\pi^3}}}$, and $\mu' = (\frac{2}{9} - \frac{1}{\varphi}cos\frac{\pi}{\varphi})\pi R_b^3 - \frac{4\pi R_a^3}{9}$.*

*Proof.* Given the unseen acquisitions of $\{q_a, q_b, q^*\}$, where $q_a \in A$, $q_b \in B$, and $q^* \in A$ or $q^* \in B$ is uncertain. However, the position of $q^*$ largely decides $h$. Therefore, the proof studies the error bounds highly related to $q^*$ in terms of two cases: $R_a \geq R_b$ and $R_a < R_b$.

1) If $R_a \geq R_b$, $q^* \in A$. Estimating the position of $q^*$ starts from the analysis on $q_a$. Given the volume function $\mathrm{Vol}(\cdot)$ over the 3-D geometry, we know: $\mathrm{Vol}(A) = \frac{4\pi}{3}R_a^3$ and $\mathrm{Vol}(B) = \frac{4\pi}{3}R_b^3$. Given $k = 3$ over $S_a$ and $S_b$, we define the minimum distance of $q_a$ to the boundary of $A$ as $d_a$. Let $S_a$ be cut off by a cross section $h'$, where $\mathcal{C}$ be the cut and $\mathcal{D}$ be the *spherical cap*[1] of the half-sphere (see Figure 12) that satisfy

$$\mathrm{Vol}(\mathcal{C}) = \frac{2}{3}\pi R_a^3 - \mathrm{Vol}(\mathcal{D}) = \frac{4\pi(R_a^3 + R_b^3)}{9}, \tag{19}$$

and the volume of $\mathcal{D}$ is

$$\mathrm{Vol}(\mathcal{D}) = \pi(\sqrt{R_a^2 - (R_a - d_a)^2})^2(R_a - d_a) + \int_0^{2\pi\sqrt{R_a^2 - (R_a - d_a)^2}} \frac{\arcsin\frac{R_a - d_a}{R_a}}{2\pi}\pi R_a^2 \, dx$$

$$= \pi(R_a^2 - (R_a - d_a)^2)(R_a - d_a) + \frac{\arcsin\frac{R_a - d_a}{R_a}}{2\pi}\pi R_a^2(2\pi\sqrt{R_a^2 - (R_a - d_a)^2}) \tag{20}$$

$$= \pi(R_a^2 - (R_a - d_a)^2)(R_a - d_a) + \pi R_a^2\arcsin\frac{R_a - d_a}{R_a}\sqrt{R_a^2 - (R_a - d_a)^2}.$$

---

[1] https://en.wikipedia.org/wiki/Spherical_cap

Let $\frac{\pi}{\varphi} = \arcsin\frac{R_a - d_a}{R_a}$, Eq. (20) can be written as

$$\mathrm{Vol}(\mathcal{D}) = \pi(R_a^2 - (R_a - d_a)^2)(R_a - d_a) + \frac{\pi R_a^3}{\varphi}cos\frac{\pi}{\varphi}. \tag{21}$$

Introducing Eq. (21) to Eq. (19), we have

$$(\frac{2}{3} - \frac{1}{\varphi}cos\frac{\pi}{\varphi})\pi R_a^3 - \pi(R_a^2 - (R_a - d_a)^2)(R_a - d_a) = \frac{4\pi(R_a^3 + R_b^3)}{9}. \tag{22}$$

Let $t = R_a - d_a$, Eq. (22) can be rewritten as

$$\pi t^3 - \pi R_a^2 t + (\frac{2}{9} - \frac{1}{\varphi}cos\frac{\pi}{\varphi})\pi R_a^3 - \frac{4\pi R_b^3}{9} = 0. \tag{23}$$

To simplify Eq. (23), let $\mu = (\frac{2}{9} - \frac{1}{\varphi}cos\frac{\pi}{\varphi})\pi R_a^3 - \frac{4\pi R_b^3}{9}$, Eq. (23) then can be written as

$$\pi t^3 - \pi R_a^2 t + \mu = 0. \tag{24}$$

The positive solution of $t$ can be

$$t = \frac{R_a^2}{3} + \sqrt[3]{-\frac{\mu}{2\pi} + \sqrt{\frac{\mu^2}{4\pi^2} - \frac{\pi^3 R_a^3}{27\pi^3}}} + \sqrt[3]{-\frac{\mu}{2\pi} - \sqrt{\frac{\mu^2}{4\pi^2} - \frac{\pi^3 R_a^3}{27\pi^3}}}. \tag{25}$$

Based on Eq. (19), we know

$$\mathrm{Vol}(\mathcal{D}) = \frac{2}{3}\pi R_a^3 - \frac{4\pi(R_a^3 + R_b^3)}{9}$$
$$= \frac{2}{9}\pi R_a^3 - \frac{4}{9}\pi R_b^3 > 0. \tag{26}$$

Thus, $\sqrt[3]{2}R_b < R_a$. We next prove $q^* \in A$. Based on Eq. (26), we know

$$\pi R_b^3 < \frac{1}{2}\pi R_a^3. \tag{27}$$

Then, the following inequalities hold: 1)$2\pi R_b^3 < \pi R_a^3$, 2) $\frac{2}{3}\pi R_b^3 < \frac{1}{3}\pi R_a^3$, and 3) $\frac{2}{3}\pi R_b^3 + \frac{1}{3}\pi R_b^3 < \frac{1}{3}\pi R_a^3 + \frac{1}{3}\pi R_b^3$. Finally, we have

$$\pi R_b^3 < \frac{1}{3}(\pi R_a^3 + \pi R_b^3). \tag{28}$$

Therefore, $\mathrm{Vol}(B) < \frac{1}{3}(\mathrm{Vol}(A) + \mathrm{Vol}(B))$. We thus know: 1) $q_a \in A$ and it is with a minimum distance $d_a$ to the boundary of $S_a$, 2) $q_b \in B$, and 3) $q^* \in A$. Therefore, class $B$ can be deemed as having a very high probability to achieve a nearly zero generalization error and the position of $q^*$ largely decides the upper bound of the generalization error of $h$.

In $S_a$ that covers class $A$, the nearly optimal error region can be bounded as the spherical cap of $S_a$ with a volume constraint of $\mathrm{Vol}(A) - \mathrm{Vol}(\mathcal{C})$. We thence have an inequality of

$$\frac{\mathrm{Vol}(A) - \mathrm{Vol}(\mathcal{C})}{\mathrm{Vol}(A) + \mathrm{Vol}(B)} < err(h, 3) < \frac{1}{3}. \tag{29}$$

We next calculate the volume of the spherical cap:

$$\mathrm{Vol}(A) - \mathrm{Vol}(\mathcal{C}) = \int_{-R_a}^{R_a - d_a} \pi x^2 d\,y$$
$$= \pi \int_{R_a - d_a}^{R_a} R_a^2 - y^2 d\,y \tag{30}$$
$$= \frac{4}{3}\pi R_a^3 - \pi d_a^2(R_a - \frac{d_a}{3}).$$

Eq. (29) then is rewritten as

$$\frac{\frac{4}{3}\pi R_a^3 - \frac{\pi}{3}(3R_a - d_a)d_a^2}{\frac{4}{3}\pi R_a^3 + \frac{4}{3}\pi R_b^3} < err(h, 3) < 0.3334, \tag{31}$$

Then, we have the error bound of

$$\frac{4R_a^3 - (3R_a - d_a)d_a^2}{4R_a^3 + 4R_b^3} < err(h, 3) < 0.3334. \tag{32}$$

Introducing $d_a = R_a - t$, Eq. (32) is written as

$$\frac{4R_a^3 - (2R_a + t)(R_a - t)^2}{4R_a^3 + 4R_b^3} < err(h, 3) < 0.3334. \tag{33}$$

2) With another assumption of $R_a < R_b$, we follow the same proof skills of $R_a \geq R_b$ and know $\frac{4R_b^3 - (3R_b - d_b)d_b^2}{4R_b^3 + 4R_a^3} < err(h, 3) < 0.3334$, i.e. where $d_b = R_b - t'$ and $t' = \frac{R_b^2}{3} + \sqrt[3]{-\frac{\mu'}{2\pi} + \sqrt{\frac{\mu'^2}{4\pi^2} - \frac{\pi^3 R_b^3}{27\pi^3}}} + \sqrt[3]{-\frac{\mu'}{2\pi} - \sqrt{\frac{\mu'^2}{4\pi^2} - \frac{\pi^3 R_b^3}{27\pi^3}}}$, and $\mu' = (\frac{2}{9} - \frac{1}{\varphi}cos\frac{\pi}{\varphi})\pi R_b^3 - \frac{4\pi R_a^3}{9}$.

We thus conclude that $\min\left\{\frac{4R_a^3 - (2R_a + t)(R_a - t)^2}{4R_a^3 + 4R_b^3}, \frac{4R_b^3 - (2R_b + t')(R_b - t')^2}{4R_b^3 + 4R_a^3}\right\} < err(h, 3) < 0.3334.$  □

## D.2 PROOF OF THEOREM 1

We next present the generalization errors against an agnostic sampling budget $k$ following the above proof technique.

*Proof.* The proof studies two cases: $R_a \geq R_b$ and $R_a < R_b$. 1) If $R_a \geq R_b$, we estimate the optimal position of $q_a$ that satisfies $q_a \in A$. Given the volume function $\text{Vol}(\cdot)$ over the 3-D geometry, we know: $\text{Vol}(A) = \frac{4\pi}{3}R_a^3$ and $\text{Vol}(B) = \frac{4\pi}{3}R_b^3$. Assume $q_a$ be the nearest representative data to the boundary of $S_a$, $q_b$ be the nearest representative data to the boundary of $S_b$, and $q^*$ be the nearest representative data to $h$ either in $S_a$ or $S_b$. Given the minimum distance of $q_a$ to the boundary of $\mathcal{A}$ as $d_a$. Let $S_a$ be cut off by a cross section $h'$, where $\mathcal{C}$ be the cut and $\mathcal{D}$ be the spherical cap of the half-sphere that satisfy

$$\text{Vol}(\mathcal{C}) = \frac{2}{3}\pi R_a^3 - \text{Vol}(\mathcal{D}) = \frac{4\pi(R_a^3 + R_b^3)}{3k}, \tag{34}$$

and the volume of $\mathcal{D}$ is

$$\text{Vol}(\mathcal{D}) = \pi(\sqrt{R_a^2 - (R_a - d_a)^2})^2(R_a - d_a) + \int_0^{2\pi\sqrt{R_a^2 - (R_a - d_a)^2}} \frac{\arcsin\frac{R_a - d_a}{R_a}}{2\pi}\pi R_a^2 \, dx$$

$$= \pi(R_a^2 - (R_a - d_a)^2)(R_a - d_a) + \frac{\arcsin\frac{R_a - d_a}{R_a}}{2\pi}\pi R_a^2(2\pi\sqrt{R_a^2 - (R_a - d_a)^2}) \tag{35}$$

$$= \pi(R_a^2 - (R_a - d_a)^2)(R_a - d_a) + \pi R_a^2\arcsin\frac{R_a - d_a}{R_a}\sqrt{R_a^2 - (R_a - d_a)^2}.$$

Let $\frac{\pi}{\varphi} = \arcsin\frac{R_a - d_a}{R_a}$, Eq. (36) can be written as

$$\text{Vol}(\mathcal{D}) = \pi(R_a^2 - (R_a - d_a)^2)(R_a - d_a) + \frac{\pi R_a^3}{\varphi}cos\frac{\pi}{\varphi}. \tag{36}$$

Introducing Eq. (36) to Eq. (34), we have

$$(\frac{2}{3} - \frac{1}{\varphi}cos\frac{\pi}{\varphi})\pi R_a^3 - \pi(R_a^2 - (R_a - d_a)^2)(R_a - d_a) = \frac{4\pi(R_a^3 + R_b^3)}{3k}. \tag{37}$$

Let $t_k = R_a - d_a$, we know

$$\pi t^3 - \pi R_a^2 t + (\frac{2k - 4}{3k} - \frac{1}{\varphi}cos\frac{\pi}{\varphi})\pi R_a^3 - \frac{4\pi R_b^3}{3k} = 0. \tag{38}$$

To simplify Eq. (38), let $\mu_k = (\frac{2k-4}{3k} - \frac{1}{\varphi}cos\frac{\pi}{\varphi})\pi R_a^3 - \frac{4\pi R_b^3}{3k}$, Eq. (38) then can be written as

$$\pi t^3 - \pi R_a^2 t + \mu_k = 0. \tag{39}$$

The positive solution of $t_k$ can be

$$t_k = \frac{R_a^2}{3} + \sqrt[3]{-\frac{\mu_k}{2\pi} + \sqrt{\frac{\mu_k^2}{4\pi^2} - \frac{\pi^3 R_a^3}{27\pi^3}}} + \sqrt[3]{-\frac{\mu_k}{2\pi} - \sqrt{\frac{\mu_k^2}{4\pi^2} - \frac{\pi^3 R_a^3}{27\pi^3}}}. \tag{40}$$

Based on Eq. (35), we know

$$
\begin{aligned}
\mathrm{Vol}(\mathcal{D}) &= \frac{2}{3}\pi R_a^3 - \frac{4\pi(R_a^3 + R_b^3)}{3k} \\
&= \frac{2k-4}{3k}\pi R_a^3 - \frac{4}{3k}\pi R_b^3 > 0.
\end{aligned}
\tag{41}
$$

Thus, $\sqrt[3]{\frac{2}{k-2}}R_b < R_a$. We next prove $q^* \in A$. According to Eq. (41), we know

$$
\pi R_b^3 < \frac{k-2}{2}\pi R_a^3.
\tag{42}
$$

Then, the following inequalities hold: 1) $\frac{2}{k-2}\pi R_b^3 < \pi R_a^3$, 2) $\frac{2}{(k-2)k}\pi R_b^3 < \frac{1}{k}\pi R_a^3$, and 3) $\frac{2}{(k-2)k}\pi R_b^3 + \frac{k^2-2k-2}{(k-2)k}\pi R_b^3 < \frac{1}{k}\pi R_a^3 + \frac{k^2-2k-2}{(k-2)k}\pi R_b^3$. Finally, we have:

$$
\begin{aligned}
\pi R_b^3 \\
&< \frac{1}{k}\pi R_a^3 + \frac{k^2-2k-2}{(k-2)k}\pi R_b^3 \\
&= \frac{1}{k}\pi(R_a^3 + R_b^2) - \frac{2}{(k-2)k}\pi R_a^3 \\
&< \frac{1}{k}\pi(R_a^3 + R_b^3).
\end{aligned}
\tag{43}
$$

Therefore, $\mathrm{Vol}(B) < \frac{1}{k}(\mathrm{Vol}(A) + \mathrm{Vol}(B))$. We thus know: 1) $q_a \in A$ and it is with a minimum distance $d_a$ to the boundary of $S_a$, 2) $q_b \in B$, and 3) $q^* \in A$. Therefore, class $B$ can be deemed as having a very high probability to achieve a zero generalization error and the position of $q^*$ largely decides the upper bound of the generalization error of $h$.

In $S_a$ that covers class $A$, the nearly optimal error region can be bounded as $\mathrm{Vol}(A) - \mathrm{Vol}(C)$. We then have the inequality of

$$
\frac{\mathrm{Vol}(A) - \mathrm{Vol}(\mathcal{C})}{\mathrm{Vol}(A) + \mathrm{Vol}(B)} < err(h,k) < \frac{1}{k}.
\tag{44}
$$

Based on the volume equation of the spherical cap in Eq. (30), we have

$$
\frac{\frac{4}{3}\pi R_a^3 - \frac{\pi}{3}(3R_a - d_a)d_a^2}{\frac{4}{3}\pi R_a^3 + \frac{4}{3}\pi R_b^3} < err(h,k) < \frac{1}{k}.
\tag{45}
$$

Then, we have the error bound of

$$
\frac{4R_a^3 - (3R_a - d_a)d_a^2}{4R_a^3 + 4R_b^3} < err(h,k) < \frac{1}{k}.
\tag{46}
$$

Introducing $d_a = R_a - t_k$, Eq. (46) is written as

$$
\frac{4R_a^3 - (2R_a + t)(R_a - t_k)^2}{4R_a^3 + 4R_b^3} < err(h,k) < \frac{1}{k}.
\tag{47}
$$

2) With another assumption of $R_a < R_b$, we follow the same proof skills of $R_a \geq R_b$ and know $\frac{4R_b^3 - (3R_b - d_b)d_b^2}{4R_b^3 + 4R_a^3} < err(h,k) < \frac{1}{k}$, where $d_b = R_b - t'_k$ and $t'_k = \frac{R_b^2}{3} + \sqrt[3]{-\frac{\mu'_k}{2\pi} + \sqrt{\frac{\mu'^2_k}{4\pi^2} - \frac{\pi^3 R_b^3}{27\pi^3}}} + \sqrt[3]{-\frac{\mu'_k}{2\pi} - \sqrt{\frac{\mu'^2_k}{4\pi^2} - \frac{\pi^3 R_b^3}{27\pi^3}}}$, and $\mu'_k = \left(\frac{2k-4}{3k} - \frac{1}{\varphi}cos\frac{\pi}{\varphi}\right)\pi R_b^3 - \frac{4\pi R_a^3}{3k}$.

We thus conclude that $\min\left\{\frac{4R_a^3 - (2R_a + t_k)(R_a - t_k)^2}{4R_a^3 + 4R_b^3}, \frac{4R_b^3(2R_b + t'_k)(R_b - t'_k)^2}{4R_b^3 + 4R_a^3}\right\} < err(h,k) < \frac{1}{k}$. $\qquad\square$

### D.3 PROOF OF THEOREM 2

*Proof.* Given class $A$ and $B$ are tightly covered by ellipsoid $E_a$ and $E_b$ in a three-dimensional geometry. Let $R_{a_1}$ be the polar radius of $E_a$, $\{R_{a_2}, R_{a_3}\}$ be the equatorial radii of $E_a$, $R_{b_1}$ be polar radius of $E_b$, and $\{R_{b_2}, R_{b_3}\}$ be the equatorial radii of $E_b$. Based on Eq. (10), we know $R_{a_i} < R_a, R_{b_i} < R_b, \forall i$, where $R_a$ and $R_b$ are the radii of the spheres over the class $A$ and $B$, respectively. We follow the same proof technique of Theorem 1 to present the generalization errors of AL with ellipsoid.

The proof studies two cases: $R_{a_1} \geq R_b$ and $R_{a_1} < R_{b_1}$. 1) If $R_{a_1} \geq R_{b_1}$, $q^* \in A$. Given the volume function $\mathrm{Vol}(\cdot)$ over the 3-D geometry, we know: $\mathrm{Vol}(A) = \frac{4\pi}{3} R_{a_1} R_{a_2} R_{a_3}$ and $\mathrm{Vol}(B) = \frac{4\pi}{3} R_{b_1} R_{b_2} R_{b_3}$. Given the minimum distance of $q_a$ to the boundary of $A$ as $d_a$. Let $E_a$ by cut off by a cross section $h'$, where $\mathcal{C}$ be the cut and $\mathcal{D}$ be the ellipsoid cap of the half-ellipsoid that satisfy

$$\mathrm{Vol}(\mathcal{C}) = \frac{2}{3}\pi R_a^1 R_a^2 R_a^3 - \mathrm{Vol}(\mathcal{D}) = \frac{4\pi(R_{a_1} R_{a_2} R_{a_3} + R_{b_1} R_{b_2} R_{b_3})}{3k}, \tag{48}$$

and the volume of $\mathcal{D}$ is approximated as

$$\mathrm{Vol}(\mathcal{D}) \approx \pi(\sqrt{R_{a_1}^2 - (R_{a_1} - d_a)^2})^2 (R_{a_1} - d_a) + \int_0^{\pi R_{a_2} R_{a_3}} \frac{arcsin\frac{R_{a_1} - d_a}{R_{a_1}}}{2\pi} \pi R_{a_1}^2 \ dx$$

$$= \pi(R_{a_1}^2 - (R_{a_1} - d_a)^2)(R_{a_1} - d_a) + \frac{arcsin\frac{R_{a_1} - d_a}{R_a}}{2\pi} \pi R_{a_1}^2 (\pi R_{a_2} R_{a_3}) \tag{49}$$

$$= \pi(R_{a_1}^2 - (R_{a_1} - d_a)^2)(R_{a_1} - d_a) + \frac{1}{2}\pi R_{a_1}^2 R_{a_2} R_{a_3} arcsin\frac{R_{a_1} - d_a}{R_{a_1}}.$$

Let $\frac{\pi}{\varphi} = arcsin\frac{R_{a_1} - d_a}{R_{a_1}}$, Eq. (49) can be written as

$$\mathrm{Vol}(\mathcal{D}) = \pi(R_{a_1}^2 - (R_{a_1} - d_a)^2)(R_{a_1} - d_a) + \frac{\pi^2}{2\varphi} R_{a_1}^2 R_{a_2} R_{a_3}. \tag{50}$$

Introducing Eq. (50) to Eq. (48), we have

$$\left(\frac{2}{3} - \frac{\pi R_{a_1}}{2\varphi}\right)\pi R_{a_1} R_{a_2} R_{a_3} - \pi(R_{a_1}^2 - (R_{a_1} - d_a)^2)(R_{a_1} - d_a) = \frac{4\pi(R_{a_1} R_{a_2} R_{a_3} + R_{b_1} R_{b_2} R_{b_3})}{3k}. \tag{51}$$

Let $\lambda_k = R_{a_1} - d_a$, we know

$$\pi\lambda_k^3 - \pi R_a^2 \lambda_k + \left(\frac{2k-4}{3k} - \frac{\pi R_{a_1}}{2\varphi}\right)\pi R_{a_1} R_{a_2} R_{a_3} - \frac{4\pi R_{b_1} R_{b_2} R_{b_3}}{3k} = 0. \tag{52}$$

To simplify Eq. (52), let $\sigma_k = \left(\frac{2k-4}{3k} - \frac{\pi R_{a_1}}{2\varphi}\right)\pi R_{a_1} R_{a_2} R_{a_3} - \frac{4\pi R_{b_1} R_{b_2} R_{b_3}}{3k}$, Eq. (52) then can be written as

$$\pi\lambda_k^3 - \pi R_{a_1}^2 \lambda_k + \sigma_k = 0. \tag{53}$$

The positive solution of $\lambda_k$ can be

$$\lambda_k = \frac{R_{a_1}^2}{3} + \sqrt[3]{-\frac{\sigma_k}{2\pi} + \sqrt{\frac{\sigma_k^2}{4\pi^2} - \frac{\pi^3 R_{a_1}^3}{27\pi^3}}} + \sqrt[3]{-\frac{\sigma_k}{2\pi} - \sqrt{\frac{\sigma_k^2}{4\pi^2} - \frac{\pi^3 R_{a_1}^3}{27\pi^3}}}. \tag{54}$$

The remaining proof process follows Eq. (40) to Eq. (46) of Theorem 1. We thus conclude that

$$\min\left\{\frac{4\prod_i R_{a_i} - (2R_{a_1} + \lambda_k)(R_{a_1} - \lambda_k)^2}{4\prod_i R_{a_i} + 4\prod_i R_{b_i}}, \frac{4\prod_i R_{b_i} - (2R_{b_1} + \lambda_k')(R_{b_1} - \lambda_k')^2}{4\prod_i R_{b_i} + 4\prod_i R_{a_i}}\right\} < err(h, k) < \frac{1}{k}, \tag{55}$$

where $i = 1, 2, 3$, $\lambda_k' = \frac{R_{b_1}^2}{3} + \sqrt[3]{-\frac{\sigma_k}{2\pi} + \sqrt{\frac{\sigma_k'^2}{4\pi^2} - \frac{\pi^3 R_{b_1}^3}{27\pi^3}}} + \sqrt[3]{-\frac{\sigma_k}{2\pi} - \sqrt{\frac{\sigma_k^2}{4\pi^2} - \frac{\pi^3 R_{a_1}^3}{27\pi^3}}}$, and $\sigma_k' = \left(\frac{2k-4}{3k} - \frac{\pi R_{b_1}}{2\varphi}\right)\pi R_{b_1} R_{b_2} R_{b_3} - \frac{4\pi R_{a_1} R_{a_2} R_{a_3}}{3k}$. In a simple way, $R_{a_1} R_{a_2} R_{a_3}$ and $R_{b_1} R_{b_2} R_{b_3}$ can be written as $\prod_i R_{a_i}$ and $\prod_i R_{b_i}$, i=1, 2, 3, respectively.

$\square$

## D.4 PROOF OF THEOREM 3

*Proof.* In clustering stability, $\gamma$-tube structure that surrounds the cluster boundary largely decides the performance of a learning algorithm. Definition of $\gamma$-tube is as follows.

**Definition 1.** $\gamma$-tube $\mathrm{Tube}_\gamma(f)$ *is a set of points distributed in the boundary of the cluster.*
$$\mathrm{Tube}_\gamma(f) := \{x \in X | \ell(x, B(f)) \leq \gamma\}, \tag{56}$$
*where $X$ is a noise-free cluster with $n$ samples, $B(f) := \{x \in X, f$ is discontinuous at $x\}$, $f$ is a clustering function, and $\ell(\cdot, \cdot)$ denotes the distance function.*

Following this conclusion, representation data can achieve the optimal generalization error if they are spread over the tube structure. Let $\gamma = d_a$, the probability of achieving a nearly zero generalization error can be expressed as the volume ration of $\gamma$-tube and $S_a$:

$$\Pr[err(h,k) = 0]_{\text{Sphere}} = \frac{\text{Vol}(\text{Tube}_\gamma)}{\text{Vol}(S_a)}$$

$$= \frac{\frac{4}{3}\pi R_a^3 - \frac{4}{3}(R_a - d_a)^3}{\frac{4}{3}\pi R_a^3} \tag{57}$$

$$= 1 - \frac{t_k^3}{R_a^3},$$

where $t_k$ keeps consistent with Eq. (40). With the initial sampling from the tube structure of class $A$, the subsequent acquisitions of AL would be updated from the tube structure of class $B$. If the initial sampling comes from the tube structure of $B$, the next acquisition must be updated from the tube structure of $A$. With the updated acquisitions spread over the tube structures of both classes, $h$ is easy to achieve a nearly zero error. Then Theorem 3 is as stated. □

### D.5 PROOF OF THEOREM 4

*Proof.* Following the proof technique of Theorem 3, volume of the tube is redefined as $\text{Vol}(\text{Tube}_\gamma) = \frac{4}{3}\pi R_{a_1} R_{a_2} R_{a_3}$. Then, we know

$$\Pr[err(h,k) = 0]_{\text{Ellipsoid}} = \frac{\text{Vol}(\text{Tube}_\gamma)}{\text{Vol}(E_a)}$$

$$= \frac{\frac{4}{3}\pi R_{a_1} R_{a_2} R_{a_3} - \frac{4}{3}(R_{a_1} - d_a)(R_{a_2} - d_a)(R_{a_3} - d_a)}{\frac{4}{3}\pi R_{a_1} R_{a_2} R_{a_3}} \tag{58}$$

$$= 1 - \frac{\lambda_{k_1}\lambda_{k_2}\lambda_{k_3}}{R_{a_1} R_{a_2} R_{a_3}},$$

where $\lambda_{k_i} = \frac{R_{a_i}^2}{3} + \sqrt[3]{-\frac{\sigma_{k_i}}{2\pi} + \sqrt{\frac{\sigma_{k_i}^2}{4\pi^2} - \frac{\pi^3 R_{a_i}^3}{27\pi^3}}} + \sqrt[3]{-\frac{\sigma_{k_i}}{2\pi} - \sqrt{\frac{\sigma_{k_i}^2}{4\pi^2} - \frac{\pi^3 R_{a_i}^3}{27\pi^3}}}$, and $\sigma_{k_i} = \left(\frac{2k-4}{3k} - \frac{\pi R_{a_i}}{2\varphi}\right)\pi R_{a_1} R_{a_2} R_{a_3} - \frac{4\pi R_{b_1} R_{b_2} R_{b_3}}{3k}$, $i = 1, 2, 3$.

Theorem 4 then is as stated. □

### D.6 PROOF OF PROPOSITION 1

*Proof.* Let $\text{Cube}_a$ tightly covers $S_a$ with a side length of $2R_a$, and $\text{Cube}'_a$ tightly covers the cut $\mathcal{C}$, following theorem 1, we know

$$err(h,k) > \frac{\text{Vol}(A) - \text{Vol}(\mathcal{C})}{\text{Vol}(A)} > \frac{\text{Cube}_a - \text{Cube}'_a}{\text{Cube}_a}. \tag{59}$$

Then, we know

$$err(h,k) > \frac{\pi R_a^3 - \pi R_a^2 d_a}{\pi R_a^3}$$

$$= 1 - \frac{d_a}{R_a}. \tag{60}$$

Meanwhile, let $\text{Cube}_e$ tightly covers $E_a$ with a side length of $2R_{a_1}$, $\text{Cube}_e'$ tightly covers $\mathcal{C}$, following theorem 1, we know

$$err(h,k) > \frac{\text{Vol}(A) - \text{Vol}(\mathcal{C})}{\text{Vol}(A)} > \frac{\text{Cube}_e - \text{Cube}'_e}{\text{Cube}_e}. \tag{61}$$

Then, we know

$$err(h,k) > \frac{\pi R_{a_1} R_{a_2} R_{a_3} - \pi d_a R_{a_2} R_{a_3}}{\pi R_{a_1} R_{a_2} R_{a_3}}$$

$$= 1 - \frac{d_a}{R_{a_1}}. \tag{62}$$

Since $R_{a_1} < R_a$, we know $1 - \frac{d_a}{R_a} > 1 - \frac{d_a}{R_{a_1}}$. It is thus the lower bound of AL with ellipsoid is tighter than AL with sphere. Then, Proposition 1 holds. □

D.7 PROOF OF PROPOSITION 2

*Proof.* Following the proofs of Theorem 3:

$$\Pr[err(h,k) = 0]_{\text{Sphere}} = 1 - \frac{t_k^3}{R_a^3}$$

$$= 1 - \frac{(R_a - d_a)^3}{R_a^3} \tag{63}$$

$$= 1 - \left(1 - \frac{d_a}{R_a}\right)^3.$$

Following the proofs Theorem 4:

$$\Pr[err(h,k) = 0]_{\text{Ellipsoid}} = 1 - \frac{\lambda_{k_1}\lambda_{k_2}\lambda_{k_3}}{R_{a_1}^3}$$

$$= 1 - \frac{(R_{a_1} - d_a)(R_{a_2} - d_a)(R_{a_3} - d_a)}{R_{a_1}^3} \tag{64}$$

$$> 1 - \left(1 - \frac{d_a}{R_{a_1}}\right)^3.$$

Based on Proposition 1, $1 - \frac{d_a}{R_a} > 1 - \frac{d_a}{R_{a_1}}$, therefore $\Pr[err(h,k) = 0]_{\text{Sphere}} < \Pr[err(h,k) = 0]_{\text{Ellipsoid}}$. Then, Proposition 2 is as stated. $\qquad\square$

D.8 LOWER DIMENSIONAL GENERALIZATION OF THE $d$-DIMENSIONAL GEOMETRY

With the above theoretical results, we next present a connection between 3-D geometry and $d$-dimensional geometry. The major technique is to prove that the volume of the 3-D geometry is a lower dimensional generalization of the $d$-dimensional geometry. It then can make all proof process from Theorems 1 to 5 hold in $d$-dimensional geometry.

**Theorem 6.** *Let* $\text{Vol}_d$ *be the volume of a $d$-dimensional geometry,* $\text{Vol}_3(S_a)$ *is a low-dimensional generalization of* $\text{Vol}_d(S_a)$.

*Proof.* Given $S_a$ over class $A$ is defined with $\mathbf{x_1^2} + \mathbf{x_2^2} + \mathbf{x_3^2} = R_a^2$. Let $\mathbf{x_1^2} + \mathbf{x_2^2} = r_a^2$ be its 2-D generalization of $S_a$, assume that $x_2$ be a variable parameter in this 2-D generalization formula, the "volume" (2-D volume is the area of the geometry object) of it can be expressed as

$$\text{Vol}_2(S_a) = \int_{-r_a}^{r_a} 2\sqrt{r_a^2 - \mathbf{x_2^2}} d\mathbf{x_2}. \tag{65}$$

Let $\vartheta$ be an angle variable that satisfies $x_2 = r_a sin(\vartheta)$, we know $d\mathbf{x_2} = r_a cos(\vartheta)d\vartheta$. Then, Eq. (65) is rewritten as

$$\text{Vol}_2(S_a) = \int_{-\pi/2}^{\pi/2} 2r_a^2 cos^2(\vartheta)d\vartheta$$

$$= \int_0^{\pi/2} 4r_a^2 cos^2(\vartheta)d\vartheta. \tag{66}$$

For a 3-D geometry, for the variable $\mathbf{x_3}$, it is over a cross-section which is a **2**-dimensional ball (circle), where the radius of the ball can be expressed as $r_a cos(\vartheta)$, s.t. $\vartheta \in [0, \pi]$. Particularly, let $\text{Vol}_3(S_a)$ be the volume of $S_a$ with 3 dimensions, the volume of this 3-dimensional sphere then can be written as

$$\text{Vol}_3(S_a) = \int_0^{\pi/2} 2\text{Vol}_2(r_a cos(\vartheta))r_a(cos(\vartheta))d\vartheta. \tag{67}$$

With Eq. (67), volume of a $d$-dimensional geometry can be expressed as the integral over the (**d**-1)-dimensional cross-section of $S_a$

$$\text{Vol}_m(S_a) = \int_0^{\pi/2} 2\text{Vol}_{m\text{-}1}(r_a cos(\vartheta))r_a(cos(\vartheta))d\vartheta, \tag{68}$$

where $\text{Vol}_{m\text{-}1}$ denotes the volume of $(m\text{-}1)$-dimensional generalization geometry of $S_a$.

Based on Eq. (68), we know $\text{Vol}_\mathbf{3}$ can be written as

$$\text{Vol}_\mathbf{3}(S_a) = \int_0^{\pi/2} 2\text{Vol}_\mathbf{2}(R_a cos(\vartheta))r_a(cos(\vartheta'))d\vartheta \tag{69}$$

Introducing Eq. (66) into Eq. (69), we have

$$\begin{aligned}
\text{Vol}_\mathbf{3}(S_a) &= \int_0^{\pi/2} 2\text{Vol}_\mathbf{2}(R_a cos(\vartheta'))r_a(cos(\vartheta'))d\vartheta' \\
&= \int_0^{\pi/2}\int_0^{\pi/2} 8(r_a cos(\vartheta')^2 cos^2(\vartheta')r_a(cos(\vartheta))d\vartheta'd\vartheta \\
&= \frac{4}{3}\pi R_a^2 \quad \text{s.t.} R_\mathrm{a} = r_\mathrm{a}.
\end{aligned} \tag{70}$$

Therefore, the generalization analysis results of the 3-D geometry still can hold in the high dimensional geometry. Then, □

**Theorem 7.** *Let* $\text{Vol}_\mathbf{d}$ *be the volume of a $\mathbf{d}$-dimensional geometry,* $\text{Vol}_\mathbf{3}(E_a)$ *is a low-dimensional generalization of* $\text{Vol}_\mathbf{d}(E_a)$*.*

*Proof.* The integral of Eq. (67) also can be adopted into the volume of $\text{Vol}_\mathbf{3}(E_a)$ by transforming the area i.e. $\text{Vol}_\mathbf{2}(S_a)$ into $\text{Vol}_\mathbf{2}(E_a)$. Then, Eq. (68) follows this transform. □

