# OpenReview forum: "On the Geometry of Deep Bayesian Active Learning"
_ICLR.cc/2021/Conference — Reject_

### Official Review · AnonReviewer4 · 2020-10-26
**An incremental contribution which seems to provide marginal improvement in performance**

**Rating:** 5
**Confidence:** 3

**Review:**

In this paper the author/s study/ies the relvant problem of acive learning, i.e. the setting in which labeled data are not available. The innovation of the paper consist in constructing the core-set on an ellipsoid, and not on the typical sphere, preventing its updates towards the boundary regions of the distributions.
The paper states the main improvements of GBALD, the proposed method, over the existing method BALD are twofold: a) to relieve the sensitivity analysis to uninformative prior
b) to reduce redundant information related to model uncertainty.
Improvements of GBALD over existing models are proved through comparison to BALD spherical interpretation.
In particular, the paper shows that a geodesic search with ellipsoid can give a "lower error bound" which is tighter than that in the specialied literature. Furthermore, the paper aims to prove that with high probability the proposed approach achieves zero error. Numerical experiments conducted under different scenarios prove GBALD improves on BALD and BatchBALD which are considered ot be state-of-the-art meyhods.

----------------------------------------------------------------------------------------------------------------------
Reason for Score:
Overall, I vote for rejecting. I like the cleaness of the proposed approach but I think that the paper in the current status is not yet mature for such a relevant venue as ICLR. However, my opinion is better described in the  Cons section.

----------------------------------------------------------------------------------------------------------------------
Pros.
1) the paper tackles a very relevant problem, the one of unlabeled data.
2) I appreciate the clean mathematical approach.
3) the proposed approach is well described.
3) numerical experiments, on the selected data sets, witness in favour of the proposed method, even if I also have concerns on this part of the paper.
----------------------------------------------------------------------------------------------------------------------
Cons.
1) a fundamental aspect of the proposed approach, when compared to alternative methods, is missing, i.e., the computational cost of the proposed approach when compared to other methods.
2) I was not able to find a discussion about the relevance or cost incidence about data labeling. Which is the concrete advantage of labeling 100 in place of 200 cases? In oother words I would like to provide motivations for the problem tackled and a quantitative assessment of the advantage of the proposed approach.
3) I'm increasingly concerned with the fact that only image datasets are taken into account in numerical experiments, images have a highly structured nature, the concept of smoothness, closeness between pixels, etc... I truly would like to see how the proposed method improves on different types of data, health data, biology data, financial data, ...
If the proposed approaches are confined to image datasets then this should be clearly stated in the paper in my humble opinion.
4) the proposed approach, even if it is well desribed and analyzed, is somewhat straightforward. Indeed, the entire contribution builds on using an ellipsoid in place of shpere.
5) In figure 4 I do not appreciate a strong difference between the proposed apporahc and BALD, at leat for digit 1 and 0, which I also would like to know why they were selected and not othe digits.
6) In figure 6, the advantage of the proposed approach over alternative methods it is limited to the very beginning of the left part of the plots, I can non understand which is the effective/practical/economic advantage of such a difference. I would kindly ask the author to help me understand it.
7) In table 1; standard deviations are huge and thus the expeted value of accuracy, achieved by differente methods, are extrelemy un certain, I would like the authors to help me understand how this impact their conclusions.
8) In Table 2, K-medoids seems to be not that bad, what about comparison in terms of computational efforts to the proposed approach?
----------------------------------------------------------------------------------------------------------------------
Questions during rebuttal period:

Please address and clarify the cons above
----------------------------------------------------------------------------------------------------------------------
Some typos:

page 2: I read "which are no longer distributed uniformly", I would like the author/s to better clarify on this
	I read "It is thus the ranking rejects unnecessary redundant acquisitions.", it makes no sense to me
page 5: equation (10), I would suggest the use of DJ and not Di, indeed in previous part you used xi and Dj, while now using xi and Di makes me confuse
----------------------------------------------------------------------------------------------------------------------

---

> ### Author Response · Authors · 2020-11-20
> **The contributions of our work can be summarized from Geometric, Algorithmic, and Theoretical perspectives.**
>
> 1. The contributions  are highlighted in Page 2.
>
>  Our Bayesian active learning from geometry  may benefit those ICLR members who are anxious about his/her deep model without reasonable theoretical explanation. Therefore, our work brings new insights for active learning with deep representation.
>
>
>
> 2. Active learning model is not easy to produce its real calculation complexity since it always iteratively calls DNNs or other classifiers.  Only the algorithms which have no interactions with the training model can have a stable calculation complexity.
>
>
> 3. Our experiments mainly compare the accuracies of different baselines under a sample acquisition size. Therefore, all learning curves are drawn to explain this issue. In Appendix C.2, C.3 and C.4 (currently Section 5.4), we also reported the accuracies of GBALD using different bath sizes.  Considering that GBALD has two main components: initial acquisitions based on core-set construction and model uncertainty estimation with those initial acquisitions,  Appendix C.6 discussed the influences of core-set size to the final performance of GBALD. From Table 6, GBALD algorithm keeps stable accuracies over the start, ultimate, and mean±std accuracies when there inputs more than 1,000 core-set labels. Therefore, drawing sufficient core-set labels using Eq. (10) to start the model uncertainty of Eq. (14) can maximize the performance of our GBALD framework.
>
>
> 4.  The current active learning community mainly focuses on the benchmark datasets to derive state-of-the-art algorithms or models. MNIST, CIFAR 10, and CIFAR100 are typical benchmark datasets to state the improvements of the proposed algorithm/model. Reviewers can see the references of Pinsler et al., 2019 and Kirsch et al., 2019.
> For other real datasets such as active labelling on MRI images, cell classification, etc, will be interesting applications. Thanks for your suggestions!
>
> “as the method partly relies on the l2 norm and it’s not a good metric for bigger images.”-> We think L_2 still is the main metric in typical image classification tasks. We here presented some computer vision papers which study active learning in bigger image datasets as ImageNet, Cityscapes, and BDD100K:
>
> Beluch, W. H., et al. The power of ensembles for active learning in image classification. CVPR 2018.
>
> Sinha, S., et al. Variational adversarial active learning. CVPR 2019
>
> Even in those computer vision application papers, CIFAR10 and CIFAR100 are still the main benchmark datasets to demonstrate the effectiveness of the proposed model. Therefore, we think our experimental settings on the dataset selections are reasonable.
>
>
> 5. This is just one our key technical innovation. We have contributions raising from Geometric, Algorithmic and Theoretical views.
>
>
> 6.  The improvements of active deep learning was not so attractive as some other deep learning applications. Reviewer can see the results of Pinsler et al., 2019 and Kirsch et al., 2019.
>
> The goal of active learning is to save the labelling cost, not improve the accuracy as typical model training. In real world, saving the labelling cost has practical values, especially for some medical images and biological DNA segments, etc.
>
> In the experimental setting of Section 5.1, selecting the labelled data from one fixed class can make the prior be uninformative. This just a random setting without any specification.
>
>
> 7.  In Figure 6, it is the test results of Appendix C.2-“Active Acquisitions with Repeated Samples”. This test aims to show the perturbations of repeated samples to the acquisition model/algorithm. Therefore, we need to observe the perturbations of the three learning curves of the model/algorithm. It is clear that Var has more significant perturbations than GBALD at the beginning of those learning curves.
>
>
>
> 8.  The results of “mean±std” in Table 1 are over the breakpoints of the learning curves in Figure 5. For example, in MNIST, we select the acquisition sizes of {0, 10, 20, 30, ..., 600}; in SVHN,  we select the acquisition sizes of {0, 100, 200, ..., 10000}; in CIFAR10, we select the acquisition sizes of {0, 100, 200, ..., 20000}.  Then, we calculate their average accuracies and std values over these acquisition points. As the shown in Table 1, all std values around 0.1, yielding a norm value. (Usually, an average accuracy on a same acquisition size with different random seeds of the deep model, will result a small std value.  Our mean accuracy spans across the whole learning curve.)  Appendix C.7 presents t-test analysis. Please see it.
>
> 9.  In the results of Table 2,  k-medoids performs not bad, but still worse than GBALD.  Usually, k-medoids has a calculation complexity of O(n^2) to O(n^3). For our GBALD, it calls the dep learning model. Therefore, it is difficult to estimate its true calculation complexity. However, the first stage of GBALD is very easy and fast. Given a large bath size, the second stage also will go fast.
>
> 10. We have updated these typos.

---

### Official Review · AnonReviewer1 · 2020-10-29
**A possibly promising but strangely written paper**

**Rating:** 4
**Confidence:** 4

**Review:**


This paper introduces a new active learning algorithm called Geometric BALD, or GBALD. While the empirical experiments look promising, the writing for this paper is a bit strange.

Comments and questions:
 - It is a bit strange to say the "deep learning community introduced active learning" and cite a 2017 paper, since active learning has been around for over 20 years before that.
 - I'm confused by the line "BALD has two interpretations: model uncertainty estimation and core-set construction" in the introduction. Can the authors provide a reference or argument for this statement? How is core-set construction an interpretation of BALD?
 - What do the authors mean by "Recently, deep Bayesian AL attracted the community."? What community?
 - How is $p(D_0 | \theta)$ a "prior"? Perhaps the authors meant $p(\theta | D_0)$, but that would be a posterior based on the initial seed set?
 - I don't think this paper uses the term "uninformative prior" correctly. For example, the paper states, "For example, unbalanced label initialization on $D_0$ usually leads to an uninformative prior,".
 - Why is the theory (appendix B) done specifically with three-dimensions? Don't most datasets have many more dimensions?
 - In appendix B, what sort of mathematical objects are A and B? They are explicitly said to be "two classes" but there are spheres that "tightly cover" A and B.

---

> ### Author Response · Authors · 2020-11-20
> **We present evidences and explanations for the misunderstanding points**
>
> 1. We here would like to state the organizations of our work and present possible assistance for the review process.
>
> Background: Active learning leverages the abundance of unlabelled data to relieve the label bottleneck of DNNs. One effective model is BALD from the view of Bayesian probabilistic interpretation, where model uncertainty estimation and core-set construction are its two generalizations.
>
> Problem Settings: Naively interacting with BALD using uninformative prior leads to unstable biased acquisitions (Gao et al., 2020). Moreover, the similarity or consistency of those acquisitions to the previous acquired samples, brings redundant information to the model and decelerates its training.
>
> Motivation: Core-set construction (Campbell & Broderick, 2018) avoids the greedy interaction to the model by capturing characteristics of the data distributions. However, data points located at the boundary regions of the distribution, usually win uniform distribution, cannot be highly-representative candidates for the core-set.
> The contributions of our work can be summarized from Geometric, Algorithmic, and Theoretical perspectives. (see updated PDF)
>
> Results: 1) theoretical results show that, compared to typical Bayesian spherical interpretation, geodesic search with ellipsoid can derive a tighter lower error bound and achieve higher probability to obtain a nearly zero error. Experiments on acquisitions with several scenarios demonstrate that, yielding slight perturbations to noisy and repeated samples, GBALD further achieves significant accuracy improvements than BALD, BatchBALD and other baselines.
>
> 2. Active learning was actually proposed around 20 years old. See a typical literature:
>
> Settles, B. (2009). Active learning literature survey.
>
> However, Bayesian active deep learning recently attracted our eyes from the work of  Gal et al., 2017. Therefore, active learning and Bayesian active deep learning are different concepts. To reduce ambiguity, we will further revise this sentence.  Thank you very much for pointing out this issue!
>
>
> 3. In the fourth paragraph of the introduction, we have presented “By modeling the complete data posterior over the distributions of parameters, BALD can be deemed as a core-set construction process on a sphere (Kirsch et al., 2019), which seamlessly solicits a compact subset to approximate the input data distribution, and efficiently mitigates the sensitivity to uninformative prior and redundant information.”  See
>
> Pinsler, Robert, et al. "Bayesian batch active learning as sparse subset approximation."  NeurIPS 2019.
>
> In Section 3.2 “Core-set Construction”, we have explained how BALD can be expressed as subset approximation. Reviewers may have overlooked something.
>
>
> 4. Active learning  had been studied very well with SVM.  Bayesian active deep learning attracted our eyes from the work of Gal et al., 2017. The community is an usual writing which means the research community such as machine learning or deep learning, etc.
>
> 5. p(D_0|θ)  is prior, and p(θ|D_0)  is posterior. See Section 3.1 and 3.2, where present the basic Bayesian rules in BALD. For example, in Section 3.1, BALD tries to maximize the decrease in expected posterior entropy; in Section 3.2, core-set construction of BALD is to approximate the log posterior of Eq. (2) by a subset.
>
>
> 6.  In wikipedia, “an uninformative prior can be created to reflect a balance among outcomes when no information is available”. See https://en.wikipedia.org/wiki/Prior_probability
>
> This is a feasible term in Bayesian statistical inference.   See more references from statistics and physics:
>
> Strachan, R.W. and Van Dijk, H.K., 2003. Bayesian model selection with an uninformative prior. Oxford Bulletin of Economics and Statistics.
>
> Price, Harold J., and Allison R. Manson. "Uninformative priors for Bayes’ theorem." American Institute of Physics, 2002.
>
>
> 7.  In our generalization analysis, we follow the perceptron learning to present the feasible error and probability analysis. Appendix D.8 presented a protocol of the 3-D geometry and d-dimensional geometry. The major technique is to prove that the volume of the 3-D geometry is a lower dimensional generalization of the d-dimensional geometry. It then can make all proof process from Theorems 1 to 5 hold in d-dimensional geometry.
>
>
>
> 8. This is the common assumption in PAC learning or perceptron training. Similar assumptions usually appears in SVM bound analysis or halfspace learning. Ball is a reasonable assumption to explore some fixed learning bounds.
> See references:
>
> Gonen, A., Sabato, S. and Shalev-Shwartz, S., 2013. Efficient active learning of halfspaces: an aggressive approach. The JMLR, 14(1), pp.2583-2615.
>
> Yan S, Zhang C. Revisiting perceptron: Efficient and label-optimal learning of halfspaces. NeurIPS  2017.

---

### Official Review · AnonReviewer2 · 2020-10-29
**Unreadable**

**Rating:** 3
**Confidence:** 4

**Review:**

This paper is basically unreadable. The sentence structure / grammar is strange, and if that was the only issue it could be overlooked. The paper also does not describe or explain the motivation and interpretation of anything, but instead just lists equations. For example, eta is the parameter that projects a spherical geodesic onto an the ellipsoid one, and an ellipsoid geodesic prevents updates of the core-set towards the boundary regions where the characteristics of the distribution cannot be captured. However, what are these characteristics, and how can they motivate how to choose eta?

---

> ### Author Response · Authors · 2020-11-20
> **It seems that the reviewer has some misunderstanding on our work**
>
> 1. Here, we would like to state the organization of our work and present possible assistance for the review process.
>
> Background: Active learning leverages the abundance of unlabelled data to relieve the label bottleneck of DNNs. One effective model is BALD from the view of Bayesian probabilistic interpretation, where model uncertainty estimation and core-set construction are its two typical generalizations.
>
> Problem Settings: Naively interacting with BALD using uninformative prior leads to unstable biased acquisitions (Gao et al., 2020). Moreover, the similarity or consistency of those acquisitions to the previous acquired samples, brings redundant information to the model and decelerates its training.
>
> Motivation: Core-set construction (Campbell & Broderick, 2018) avoids the greedy interaction to the model by capturing characteristics of the data distributions. However, data points located at the boundary regions of the distribution, usually win uniform distribution, cannot be highly-representative candidates for the core-set.
>
> The contributions of our work can be summarized from Geometric, Algorithmic, and Theoretical perspectives: 1) Geometrically, our key innovation is to construct the core-set on an ellipsoid, not typical sphere, preventing its updates towards the boundary regions of the distributions; 2) In term of algorithm design, our work proposes a two-stage framework from a Bayesian perspective that sequentially introduces the core-set representation and model uncertainty, strengthening their performance “independently”. Moreover, different to the typical BALD optimizations, we present geometric solvers to construct core-set and estimate model uncertainty, which result in a different view for Bayesian active learning; 3) Theoretically, to guarantee those improvements, our generalization analysis proves that, compared to typical Bayesian spherical interpretation, geodesic search with ellipsoid can derive a tighter lower error bound and achieve higher probability to obtain a nearly zero error.
>
>
> Results: Our theoretical results show that, compared to typical Bayesian spherical interpretation, geodesic search with ellipsoid can derive a tighter lower error bound and achieve higher probability to obtain a nearly zero error. Experiments on acquisitions with several scenarios demonstrate that, yielding slight perturbations to noisy and repeated samples, GBALD further achieves significant accuracy improvements over BALD, BatchBALD and other baselines.
>
> We have clearly presented mathematic background and each equation is properly presented and explained. I don’t understand why the reviewer complains that our math is not clear. Based on the comment of “Good mathematical justification for the method” from AnonReviewer3, and “I appreciate the clean mathematical approach” from AnonReviewer4, our math contents should be readable.
>
> I suggest the reviewer should reconsider the score after carefully reading our main text and appendix.
>
>
>
> 2. In core-set, data in boundary regions cannot be an expected selection due to its low-representativeness. Therefore, the “characteristics” means the representativeness of one data. We present clear parameter definitions on $\eta$ that rescale the spherical geodesic into ellipsoid geodesic. Appendix presents detailed proofs for their error bounds and the probability to obtain a nearly zero error. In Appendix C.1, we present our experimental setting on η. In Appendix C.5, we discussed the parameter settings: “Ellipsoid geodesic is adjusted by η which controls how far of the updates of core-set to the boundaries of the distributions. It is set as η = 0.9 in this paper.”.
>
> From the both theoretical and algorithmic aspects, we have explained the parameter setting as clear as possible. It seems that the reviewer  may have some misunderstanding on our work.

---

### Official Review · AnonReviewer3 · 2020-10-31
**Interesting idea and well-designed method, but experimental results and baselines are relatively weak**

**Rating:** 5
**Confidence:** 5

**Review:**

Quality and clarity: the paper is well-written with good knowledge of the current state of the research in the subject area.

Significance: active learning is an important practical direction in machine learning. It currently has two main approaches - one is based on the uncertainty and another one on selecting the most representative subsets. The effective combination of these two is a promising direction of the research and it is addressed in the paper.

Originality: The idea to combine the uncertainty and diverse points selection is not exactly novel, but the paper proposes an interesting idea of improving the point selection with the rescaled geodesic search.

Pros:
* The proposed method is well-reasoned
* Good mathematical justification for the method
* Tractable method implementation is presented.

Cons:
* The results in Table 1 are a bit confusing as the provided std values are quite large. The explicit p-value for the proposed method could be relevant.
* The experiments compare the new method only with basic ones like k-centers and BALD. More complex approaches (like batch BALD [Kirsch, 2019] and Sparse Subset Approximation [Pinsler, 2019]) were mentioned, but not presented in the main paper experiments for some reason, while they use a similar idea and show comparable improvement on the image datasets.
* Experiments 5.1, 5.2 - with a small number of initial samples and large batches, the BALD sometimes performs worse than random sampling; it would be good to add random sampling here for reference.
* The experiments are on the relatively small and simple image datasets, but it’s not clear how well the method would work on real-world size images, as the method partly relies on the l2 norm and it’s not a good metric for bigger images.

Reference
Andreas Kirsch, Joost van Amersfoort, and Yarin Gal. Batchbald: Efficient and diverse batch
acquisition for deep bayesian active learning. In Advances in Neural Information Processing
Systems, pp. 7024–7035, 2019.

Robert Pinsler, Jonathan Gordon, Eric Nalisnick, and José Miguel Hernández-Lobato. Bayesian batch
active learning as sparse subset approximation. In Advances in Neural Information Processing
Systems, pp. 6356–6367, 2019.

---

> ### Author Response · Authors · 2020-11-20
> **Reviewer may have overlooked the discussion of BatchBALD in appendix**
>
> 1. Thank you very much for your recognition of the significance of this paper.
>
> 2. In active learning, uncertainty and diversity can be optimized within one unified acquisition function using either maximization or minimization.  However, it is non-trivial to find those data satisfying the both characteristics.  Therefore, the acquisitions tend to be biased towards one of these two criteria. This paper lodges a unified deep Bayesian active learning framework. The contributions of our work can be summarized from geometric, algorithmic, and theoretical perspectives.
>
> Geometrically, our key innovation is to construct the core-set on an ellipsoid, not typical sphere, preventing its updates towards the boundary regions of the distributions.
>
> In term of algorithm design, our work proposes a two-stage framework from a Bayesian perspective that sequentially introduces the core-set representation and model uncertainty, strengthening their performance “independently”.  Moreover, different to the typical BALD optimizations, we present geometric solvers to construct core-set and estimate model uncertainty, which result in a different view for Bayesian active learning.
>
> Theoretically, to guarantee those improvements, our generalization analysis proves that, compared to typical Bayesian spherical interpretation, geodesic search with ellipsoid can derive a tighter lower error bound and achieve higher probability to obtain a nearly zero error.
>
> 3. The results of “mean±std” in Table 1 are  computed over the breakpoints of the learning curves in Figure 5. For example, in MNIST, we select the acquisition sizes of {0, 10, 20, 30, ..., 600}; in SVHN,  we select the acquisition sizes of {0, 100, 200, ..., 10000}; in CIFAR10, we select the acquisition sizes of {0, 100, 200, ..., 20000}.  Then, we calculate their average accuracies and std values over these acquisition points. As the shown in Table 1, all std values around 0.1, yielding a norm value. (Usually, an average accuracy on the same acquisition size with different random seeds of DNNs, will result in a small std value.  Our mean accuracy spans across the whole learning curve.) Appendix C.7 presents t-test analysis. Please see it.
>
> 4. Reviewer may have overlooked the appendix. In Appendix C.1-Baselines, we introduced BatchBALD w.r.t. Eq.~(18).  Experiments of C.4 (currently Section 5.4) discussed BatchBALD.
>
> For another Bayesian active learning work “Bayesian Batch Active Learning as Sparse Subset Approximation”, it is similar to the work of k-centers, which cannot be called the deep learning model. Therefore, it has disagreements on how to fairly compare those representative sampling approaches to those model uncertainty ones.  In our current experimental settings, we didn’t further study more representative sampling works because they doesn’t relate to the deep learning model.
>
>
> 5. Yes. True! As the discussion of C.2 of Appendix, BatchBALD degenerates the test accuracies if we progressively increase the batch sizes, where BatchBALD using a batch size of 10 keeps similar learning curves as BALD.  From a theoretical view, the larger the batch size, the worse the batch acquisitions will be.  Therefore, BALD and BatchBALD may have worse performance if we set an improper batch size; sometimes may be worse than random sampling. This point has been already discussed in the work of Kirsch et al., 2019. We have added the learning curves of random sampling in Figure~3. It is clear that BALD is sensitive to the uninformative prior, which also is the reason why we utilize a core-set to start BALD in future.
>
> 6.  The current active learning community mainly focuses on the benchmark datasets to derive state-of-the-art algorithm or model. MNIST, CIFAR 10, and CIFAR100 are typical benchmark datasets to state the improvements of the proposed algorithm/model. Reviewers can check the references of Pinsler et al., 2019 and Kirsch et al., 2019.
> For other real datasets such as active labelling on MRI images, cell classification, etc, will be interesting applications. Thanks for your suggestions!
>
> “as the method partly relies on the l2 norm and it’s not a good metric for bigger images.”->  L_2 is still the main metric in typical image classification tasks of active learning. We here list some computer vision papers, which study active learning in bigger image datasets as ImageNet, Cityscapes, and BDD100K, etc:
>
> Beluch, W. H., Genewein, T., Nürnberger, A., & Köhler, J. M. The power of ensembles for active learning in image classification. CVPR 2018.
>
> Sinha, S., Ebrahimi, S. and Darrell, T.. Variational adversarial active learning. CVPR 2019.
>
> Even in these latest application papers, CIFAR10 and CIFAR100 are still the main benchmark datasets to demonstrate the effectiveness of the proposed models. Therefore, we believe our dataset selections are reasonable.

---

> > ### Author Response · Authors · 2020-11-22
> > **Two-sided $t$-test**
> >
> > C.7 two-sided  $t$-test
> >
> > We present two-sided (two-tailed) $t$-test for the learning curves of Figure 5.   Different to the mean$\pm$ std of Table~1, $t$-test can enlarge the significant difference of those baselines.  In typical $t$-test, the two groups of observations usually require a degree of freedom  smaller than 30. However, the numbers of  breakpoints of MNIST, SVHN, and CIFAR10 are 61, 101, and 201, respectively, thereby holding a degree of freedom of 60, 100, 200, respectively. It is thus we introduce $t$-test score to directly compare the significant difference of pairwise baselines.
> >
> >
> > In two-sided $t$-test,  $B_1$ beats $B_2$ on breakpoints $\alpha_i$ and $\beta_i$ satisfying a condition of  $t-{\rm score}>\nu$; $B_2$ beats $B_1$ on breakpoints $\alpha_i$ and $\beta_i$ satisfying  a condition of  $t-{\rm score}<-\nu$, where $\nu$ denotes the hypothesized criterion with a given confidence risk. Following \citep{ash2019deep}, we add a penalty of $\frac{1}{e}$  to each pair of breakpoints, which further enlarges their differences in the aggregated penalty matrix, where $e$ denotes the number of  $B_1$ beats $B_2$ on all breakpoints.    All penalty values finally calculate their  $L_1$ expressions.
> >
> >
> > Figure 12 presents the penalty matrix over learning curves of Figure~5.  Column-wise values
> > at the bottom of each matrix show the overall performance of the compared baselines. As the shown results, GBALD has significant performances than that of the other baselines over the three datasets. Especially for SVHN, it has superior performance.
> >
> > We summarize the  column-wise values at the bottom of each penalty matrix.
> >
> >
> >
> >            Var|    BALD  |  Entropy |  k-medoids|  k-centers|  GBALD
> >
> > MNIST  | 2.5053    |     3.0837   |  $\quad $     4.8024     $\quad $   |    $\quad $      0.5782     |         $\qquad $              0.8843    |        $\quad $     18.7898
> >
> > SVHN   |  $\ $ 0.7556      |   2.3351     |  $\quad $  0.7023     $\quad $    |  $\quad $ 7.4051       |     $\qquad $        0.9958  | $\quad $     77.2318
> >
> > CIFAR10  | 2.721   |    1.1112       |  $\quad $ 4.3284   $\quad $ | $\quad $      4.4366      |     $\quad $   $\quad $       0.4728  |     $\quad $  9.9326

---

### Author Response · Authors · 2020-11-22
**We highlighted  our contributions from geometric, algorithmic, and theoretical perspectives.**


1. Geometrically, our key innovation is to construct the core-set on an ellipsoid, not typical sphere, preventing its updates towards the boundary regions of the distributions.



2. In term of algorithm design, our work proposes a two-stage framework from a Bayesian perspective that sequentially introduces the core-set representation and model uncertainty, strengthening their performance “independently”. Moreover, different to the typical BALD optimizations, we present geometric solvers to construct core-set and estimate model uncertainty, which result in a different view for Bayesian active learning.



3. Theoretically, to guarantee those improvements, our generalization analysis proves that, compared to typical Bayesian spherical interpretation, geodesic search with ellipsoid can derive a tighter lower error bound and achieve higher probability to obtain a nearly zero error. See Appendix B.

---

### Decision · Program_Chairs · 2021-01-07
**Final Decision**

**Decision:**

Reject

**Comment:**

All reviewers express concerns, such as about the presentation, the situation of the paper w.r.t. prior work, the experimental evaluation etc., and recommend rejection.